# Estimating Koopman operators with sketching to provably learn large scale dynamical systems

**Giacomo Meanti***
Istituto Italiano di Tecnologia
giacomo.meanti@iit.it

**Antoine Chatalic***
Università di Genova
antoine.chatalic@dibris.unige.it

**Vladimir R. Kostic**
Istituto Italiano di Tecnologia
University of Novi Sad
vladimir.kostic@iit.it

**Pietro Novelli**
Istituto Italiano di Tecnologia
pietro.novelli@iit.it

**Massimiliano Pontil**
Istituto Italiano di Tecnologia
University College London
massimiliano.pontil@iit.it

**Lorenzo Rosasco**
Università di Genova
Istituto Italiano di Tecnologia
Massachusetts Institute of Technology
lrosasco@mit.edu

## Abstract

The theory of Koopman operators allows to deploy non-parametric machine learning algorithms to predict and analyze complex dynamical systems. Estimators such as principal component regression (PCR) or reduced rank regression (RRR) in kernel spaces can be shown to provably learn Koopman operators from finite empirical observations of the system's time evolution. Scaling these approaches to very long trajectories is a challenge and requires introducing suitable approximations to make computations feasible. In this paper, we boost the efficiency of different kernel-based Koopman operator estimators using random projections (sketching). We derive, implement and test the new "sketched" estimators with extensive experiments on synthetic and large-scale molecular dynamics datasets. Further, we establish non asymptotic error bounds giving a sharp characterization of the trade-offs between statistical learning rates and computational efficiency. Our empirical and theoretical analysis shows that the proposed estimators provide a sound and efficient way to learn large scale dynamical systems. In particular our experiments indicate that the proposed estimators retain the same accuracy of PCR or RRR, while being much faster. Code is available at `https://github.com/Giodiro/NystromKoopman`.

## 1 Introduction

In the physical world, temporally varying phenomena are everywhere, from biological processes in the cell to fluid dynamics to electrical fields. Correspondingly, they generate large amounts of data both through experiments and simulations. This data is often analyzed in the framework of dynamical systems, where the state of a system $\boldsymbol{x}$ is observed at a certain time $t$, and the dynamics is described by a function $f$ which captures its evolution in time

$$\boldsymbol{x}_{t+1} = f(\boldsymbol{x}_t).$$

---

*Equal contribution

37th Conference on Neural Information Processing Systems (NeurIPS 2023).

The function $f$ must capture the whole dynamics, and as such it may be non-linear and even stochastic for instance when modeling stochastic differential equations, or simply noisy processes. Applications of this general formulation arise in fields ranging from robotics, atomistic simulations, epidemiology, and many more. Along with a recent increase in the availability of simulated data, data-driven techniques for learning the dynamics underlying physical systems have become commonplace. The typical approach of such techniques is to acquire a dataset of training pairs $(\boldsymbol{x}_t, \boldsymbol{y}_t = \boldsymbol{x}_{t+1})$ sampled in time, and use them to learn a model for $f$ which minimizes a forecasting error. Since dynamical systems stem from real physical processes, forecasting is not the only goal and the ability to interpret the dynamics is paramount. One particularly important dimension for interpretation is the separation of dynamics into multiple temporal scales: fast fluctuations can e.g. be due to thermodynamical noise or electrical components in the system, while slow dynamics describe important conformational changes in molecules or mechanical effects.

Koopman operator theory [27, 28] provides an elegant framework in which the potentially non-linear dynamics of the system can be studied via the Koopman operator

$$(\mathcal{K}\psi)(\boldsymbol{x}) = \mathbf{E}\big[\psi(f(\boldsymbol{x}))\big], \tag{1}$$

which has the main advantage of being linear but is defined on a typically infinite-dimensional set of observable functions. The expectation in (1) is taken with respect to the potential stochasticity of $f$. Thanks to its linearity, the operator $\mathcal{K}$ can e.g. be applied twice to get two-steps-ahead forecasts, and one can compute its spectrum (beware however that $\mathcal{K}$ is not self-adjoint, unless the dynamical process is time-reversible). Accurately approximating the Koopman operator and its spectral properties is of high interest for the practical analysis of dynamical systems. However doing so efficiently for long temporal trajectories remains challenging. In this paper we are interested in designing estimators which are both theoretically accurate and computationally efficient.

**Related works**   Learning the spectral properties of the Koopman operator directly from data has been considered for at least three decades [39, 40], resulting in a large body of previous work. Among the different approaches proposed over time (see Mezić [41] for a recent review) it is most common to search for finite dimensional approximations to the operator, from which part of the spectrum and the Koopman modes [8] can be obtained. Dynamic mode decomposition (DMD) [56, 63], time-lagged independent component analysis (tICA) [42, 49] and many subsequent extensions [31] for example can be seen as minimizers of the forecasting error when $\psi$ is restricted to be a linear function of the states [52]. Extended DMD (eDMD) [66, 25] and the variational approach for conformation dynamics (VAC) [45, 46] instead allow for a (potentially learnable, as in recent deep learning algorithms [32, 37, 69, 62]) dictionary of non-linear functions $\psi$. Kernel DMD [67, 26] and kernel tICA [57] are further generalizations which again approximate the Koopman operator but using an infinite dimensional space of features $\psi$, encoded by the feature map of a reproducing kernel. While often slow from a computational point of view, kernel methods are highly expressive and can be analyzed theoretically, to prove convergence and derive learning rates of the resulting estimators [29, 30]. Approximate kernel methods which are much faster to run have been recently used for Koopman operator learning by Baddoo et al. [6] where an iterative procedure is used to identify the best approximation to the full kernel, but no formal learning rates are demonstrated, and by Ahmad et al. [3] who derive learning rates in Hilbert-Schmidt norm (while we consider operator norm) for the Nyström KRR estimator (one of the three considered in this paper).

**Contributions**   In this paper we adopt the kernel learning approach. Starting from the problem of approximating the Koopman operator in a reproducing kernel Hilbert space, we derive three different estimators based on different inductive biases: kernel ridge regression (KRR) which comes from Tikhonov regularization, principal component regression (PCR) which is equivalent to DMD and its extensions, and reduced rank regression (RRR) which comes from a constraint on the maximum rank of the estimator [24]. We show how to overcome the computational scalability problems inherent in full kernel methods using an approximation based on random projections which is known as the Nyström method [58, 65]. The approximate learning algorithms scale very easily to the largest datasets, with a computational complexity which goes from $O(n^3)$ for the exact algorithm to $O(n^2)$ for the approximate one. We can further show that the Nyström KRR, PCR and RRR estimators have the same convergence rates as their exact, slow counterparts – which are known to be optimal under our assumptions. We provide learning bounds in operator norm, which are known to translate to bounds for dynamic mode decomposition and are thus of paramount importance for applications. Finally, we thoroughly validate the approximate PCR and RRR estimators on synthetic dynamical systems, comparing efficiency and accuracy against their exact counterparts [29], as well as recently

proposed fast Koopman estimator streaming KAF [20]. To showcase a realistic scenario, we train on a molecular dynamics simulation of the fast-folding Trp-cage protein [35].

**Structure of the paper**   We introduce the setting in Section 2, and define our three estimators in Section 3. In Section 4 we provide bounds on the excess risk of our estimators, and extensive experiments on synthetic as well as large-scale molecular dynamics datasets in Section 5.

## 2   Background and related work

**Notation**   We consider a measurable space $(\mathcal{X}, \mathcal{B})$ where $\mathcal{X}$ corresponds to the state space, and denote $L^2_\pi := L^2(\mathcal{X}, \mathcal{B}, \pi)$ the $L^2$ space of functions on $\mathcal{X}$ w.r.t. to a probability measure $\pi$, and $L^\infty_\pi$ the space of measurable functions bounded almost everywhere. We denote $\mathrm{HS}(\mathcal{H})$ the space of Hilbert-Schmidt operators on a space $\mathcal{H}$.

**Setting**   The setting we will consider is that of Markovian, time-homogeneous stochastic process $\{X_t\}_{t \in \mathbb{N}}$ on $\mathcal{X}$. By definition of a Markov process, $X_t$ only depends on $X_{t-1}$ and not on any previous states. Time-homogeneity ensures that the transition probability $\mathbb{P}[X_{t+1} \in B | X_t = \boldsymbol{x}]$ for any measurable set $B$ does not depend on $t$, and can be denoted with $p(\boldsymbol{x}, B)$. This implies in particular that the distribution of $(X_t, X_{t+1})$ does not depend on $t$, and we denote it $\rho$ in the following. We further assume the existence of the *invariant* density $\pi$ which satisfies $\pi(B) = \int_{\mathcal{X}} \pi(\boldsymbol{x}) p(\boldsymbol{x}, B) \, \mathrm{d}\boldsymbol{x}$. This classical assumption allows one to study a large class of stochastic dynamical systems, but also deterministic systems on the attractor, see e.g. [14]. The Koopman operator $\mathcal{K}_\pi : L^2_\pi(\mathcal{X}) \to L^2_\pi(\mathcal{X})$ is a bounded linear operator, defined by

$$(\mathcal{K}_\pi g)(\boldsymbol{x}) = \int_{\mathcal{X}} p(\boldsymbol{x}, \boldsymbol{y}) g(\boldsymbol{y}) \, \mathrm{d}\boldsymbol{y} = \mathbf{E}[g(X_{t+1})|X_t = \boldsymbol{x}], \quad g \in L^2_\pi(\mathcal{X}), \boldsymbol{x} \in \mathcal{X}. \tag{2}$$

We are in particular interested in the eigenpairs $(\lambda_i, \varphi_i) \in \mathbb{C} \times L^2_\pi$, that satisfy

$$\mathcal{K}_\pi \varphi_i = \lambda_i \varphi_i. \tag{3}$$

Through this decomposition it is possible to interpret the system by separating fast and slow processes, or projecting the states onto fewer dimensions [15, 19, 7]. In particular, the Koopman mode decomposition (KMD) allows to propagate the system state in time. Given an observable $g : \mathcal{X} \to \mathbb{R}^d$ such that $g \in \mathrm{span}\{\varphi_i | i \in \mathbb{N}\}$, the modes allow to reconstruct $g(\boldsymbol{x})$ with a Koopman eigenfunction basis. The modes $\boldsymbol{\eta}_i^g \in \mathbb{C}^d$ are the coefficients of this basis expansion:

$$(\mathcal{K}_\pi g)(\boldsymbol{x}) = \mathbf{E}[g(X_t)|X_0 = \boldsymbol{x}] = \sum_i \lambda_i \varphi_i(\boldsymbol{x}) \boldsymbol{\eta}_i^g. \tag{4}$$

This decomposition describes the system's dynamics in terms of a stationary component (the Koopman modes), a temporal component (the eigenvalues $\lambda_i$) and a spatial component (eigenfunctions $\varphi_i$).

**Kernel-based learning**   In this paper we approximate $\mathcal{K}_\pi$ with kernel-based algorithms, using operators in reproducing kernel Hilbert spaces (RKHS) $\mathcal{H}$ associated with kernel $k : \mathcal{X} \times \mathcal{X} \to \mathbb{R}$ and feature map $\phi : \mathcal{X} \to \mathcal{H}$. We wish to find an operator $A : \mathcal{H} \to \mathcal{H}$ which minimizes the risk

$$\mathcal{R}_{\mathrm{HS}}(A) = \mathbf{E}_\rho[\ell(A, (\boldsymbol{x}, \boldsymbol{y}))] \quad \text{where} \quad \ell(A, (\boldsymbol{x}, \boldsymbol{y})) := \|\phi(\boldsymbol{y}) - A\phi(\boldsymbol{x})\|^2. \tag{5}$$

The adjoint of $A$, denoted by $A^*$, should thus be understood as an estimator of the Koopman operator $\mathcal{K}_\pi$ in $\mathcal{H}$ as will be clarified in (15). In practice $\pi$ and $\rho$ are unknown, and one typically has access to a dataset $\{(\boldsymbol{x}_i, \boldsymbol{y}_i)\}_{i=1}^n$ sampled from $\rho$, where each pair $(\boldsymbol{x}_i, \boldsymbol{y}_i = f(\boldsymbol{x}_i))$ may equivalently come from a single long trajectory or multiple shorter ones concatenated together. We thus use the empirical risk

$$\hat{\mathcal{R}}_{\mathrm{HS}}(A) = \frac{1}{n} \sum_{i=1}^n \ell(A, (\boldsymbol{x}_i, \boldsymbol{y}_i)) \tag{6}$$

as a proxy for (5). Since minimizing eq. (6) may require finding the solution to a very badly conditioned linear system, different regularization methods (such as Tikhonov or truncated SVD) can be applied on top of the empirical risk.

**Remark 2.1 (Connections to other learning problems):** *The problem of minimizing eqs.* (5) *and* (6) *has strong connections to learning conditional mean embeddings [59, 22, 44, 33] where the predictors and targets are embedded in different RKHSs, and to structured prediction [12, 13] which is an even more general framework. On the other hand, the most substantial difference from the usual kernel regression setting [9] is the embedding of both targets and predictors into a RKHS, instead of just targets.*

We denote the input and cross covariance $C = \mathbf{E}_\pi[\phi(\boldsymbol{x}) \otimes \phi(\boldsymbol{x})]$ and $C_{YX} = \mathbf{E}_\rho[\phi(\boldsymbol{y}) \otimes \phi(\boldsymbol{x})]$, and their empirical counterparts as $\hat{C} = \frac{1}{n}\sum_{i=1}^{n}[\phi(\boldsymbol{x}_i) \otimes \phi(\boldsymbol{x}_i)]$ and $\hat{C}_{YX} = \frac{1}{n}\sum_{i=1}^{n}\phi(\boldsymbol{y}_i) \otimes \phi(\boldsymbol{x}_i)$. We also use the abbreviation $C_\lambda := C + \lambda I$. Minimizing the empirical risk (6) with Tikhonov regularization [9] yields the following KRR estimator

$$\hat{A}_\lambda = \underset{A \in \mathrm{HS}(\mathcal{H})}{\arg\min} \hat{\mathcal{R}}_{\mathrm{HS}}(A) + \lambda\|A\|_{\mathrm{HS}}^2 = \hat{C}_{YX}(\hat{C} + \lambda I)^{-1}. \tag{7}$$

Eq. (7) can be computed by transforming its expression with the kernel trick [23], to arrive at a form where one must invert the kernel matrix – a $n \times n$ matrix whose $i, j$-th entry is $k(\boldsymbol{x}_i, \boldsymbol{x}_j)$. This operation requires $O(n^3)$ time and $O(n^2)$ memory, severely limiting the scalability of KRR to $n \lesssim 100\,000$ points. Improving the scalability of kernel methods is a well-researched topic, with the most important solutions being random features [50, 51, 68, 21] and random projections [58, 65, 21]. In this paper we use the latter approach, whereby the kernel matrix is assumed to be approximately low-rank and is *sketched* to a lower dimensionality. In particular we will use the Nyström method to approximate the kernel matrix projecting it onto a small set of inducing points, chosen among the training set. The sketched estimators are much more efficient than the exact ones, increasingly so as the training trajectories become longer. For example, the state of the art complexity for solving (non vector valued) approximate kernel ridge regression is $O(n\sqrt{n})$ time instead of $O(n^3)$ [38, 1, 10]. Furthermore, when enough inducing points are used (typically on the order of $\sqrt{n}$), the learning rates of the exact and approximate estimators are the same, and optimal [5, 53]. Hence it is possible – and in this paper we show it for learning the Koopman operator – to obtain large efficiency gains, without losing anything in terms of theoretical guarantees of convergence.

## 3 Nyström estimators for Koopman operator regression

In this section, we introduce three efficient approximations of the KRR, PCR and RRR estimators of the Koopman operator. Our estimators rely on the Nyström approximation, i.e. on random projections onto low-dimensional subspaces of $\mathcal{H}$ spanned by the feature-embeddings of subsets of the data. We thus consider two sets of $m \ll n$ inducing points $\{\tilde{\boldsymbol{x}}_j\}_{j=1}^m \subset \{\boldsymbol{x}_t\}_{t=1}^n$ and $\{\tilde{\boldsymbol{y}}_j\}_{j=1}^m \subset \{\boldsymbol{y}_t\}_{t=1}^n$ sampled respectively from the input and output data. The choice of these inducing points (also sometimes called Nyström centers) is important to obtain a good approximation. Common choices include uniform sampling, leverage score sampling [17, 55], and iterative procedures such as the one used in [6] to identify the most relevant centers. In this paper we focus on uniform sampling for simplicity, but we stress that our theoretical results in Section 4 can easily be extended to leverage scores sampling by means of [53, Lemma 7]. To formalize the Nyström estimators, we define operators $\widetilde{\Phi}_X, \widetilde{\Phi}_Y : \mathbb{R}^m \to \mathcal{H}$ as $\widetilde{\Phi}_X w = \sum_{j=1}^m w_j\phi(\tilde{\boldsymbol{x}}_j)$ and $\widetilde{\Phi}_Y w = \sum_{j=1}^m w_j\phi(\tilde{\boldsymbol{y}}_j)$, and denote $P_X$ and $P_Y$ the orthogonal projections onto $\operatorname{span}\widetilde{\Phi}_X$ and $\operatorname{span}\widetilde{\Phi}_Y$ respectively.

In the following paragraphs we apply the projection operators to three estimators corresponding to different choices of regularization. For each of them a specific proposition (proven in Section C) states an efficient way of computing it based on the kernel trick. For this purpose we introduce the kernel matrices $K_{\tilde{X},X}, K_{\tilde{Y},Y} \in \mathbb{R}^{m \times n}$ between training set and inducing points with entries $(K_{\tilde{X},X})_{ji} = k(\tilde{\boldsymbol{x}}_j, \boldsymbol{x}_i)$, $(K_{\tilde{Y},Y})_{ji} = k(\tilde{\boldsymbol{y}}_j, \boldsymbol{y}_i)$, and the kernel matrices of the inducing points $K_{\tilde{X},\tilde{X}}, K_{\tilde{Y},\tilde{Y}} \in \mathbb{R}^{m \times m}$ with entries $(K_{\tilde{X},X})_{jk} = k(\tilde{\boldsymbol{x}}_j, \tilde{\boldsymbol{x}}_k)$ and $(K_{\tilde{X},X})_{jk} = k(\tilde{\boldsymbol{y}}_j, \tilde{\boldsymbol{y}}_k)$.

**Kernel Ridge Regression (KRR)** The cost of computing $\hat{A}_\lambda$ defined in Eq. (7) is $O(n^3)$ [29] which is prohibitive for datasets containing long trajectories. However, applying the projection operators to each side of the empirical covariance operators, we obtain an estimator which additionally depends on the $m$ inducing points:

$$\hat{A}_{m,\lambda}^{\mathrm{KRR}} := P_Y \hat{C}_{YX} P_X (P_X \hat{C} P_X + \lambda I)^{-1} : \mathcal{H} \to \mathcal{H}. \tag{8}$$

If $\mathcal{H}$ is infinite dimensional, Eq. (8) cannot be computed directly. Proposition 3.1 (proven in Section C) provides a computable version of the estimator.

**Proposition 3.1 (Nyström KRR):** *The Nyström KRR estimator* (8) *can be expressed as*

$$\hat{A}_{m,\lambda}^{KRR} = \widetilde{\Phi}_Y K_{\tilde{Y},\tilde{Y}}^\dagger K_{\tilde{Y},Y} K_{X,\tilde{X}} (K_{\tilde{X},X} K_{X,\tilde{X}} + n\lambda K_{\tilde{X},\tilde{X}})^\dagger \widetilde{\Phi}_X^*. \tag{9}$$

*The computational bottlenecks are the inversion of an $m \times m$ matrix and a large matrix multiplication, which overall need $O(2m^3 + 2m^2 n)$ operations. In particular, in Section 4 we will show that $m \asymp \sqrt{n}$ is sufficient to guarantee optimal rates even with minimal assumptions, leading to a final cost of $O(n^2)$. Note that a similar estimator was derived in [3].*

Please note that the $O(n^2)$ cost is for a straightforward implementation, and can indeed be reduced via iterative linear solvers (possibly preconditioned, to further reduce the practical running time), and randomized linear algebra techniques. In particular, we could leverage results from Rudi et al. [54] to reduce the computational cost to $O(n\sqrt{n})$.

**Principal Component Regression (PCR)**    Typical settings in which Koopman operator theory is used focus on the decomposition of a dynamical system into a small set of components, obtained from the eigendecomposition of the operator itself. For this reason, a good prior on the Koopman estimator is for it to be low rank. The kernel PCR estimator $\hat{A}^{\mathrm{PCR}} = \hat{C}_{YX}[\![\hat{C}]\!]_r^\dagger$ formalizes this concept [29, 67], where here $[\![\cdot]\!]_r$ denotes the truncation to the first $r$ components of the spectrum. Again this is expensive to compute when $n$ is large, but the estimator can be sketched as follows:

$$\hat{A}_m^{\mathrm{PCR}} = P_Y \hat{C}_{YX} [\![P_X \hat{C} P_X]\!]_r^\dagger. \tag{10}$$

The next proposition provides an efficiently implementable version of this estimator.

**Proposition 3.2 (Nyström PCR):** *The sketched PCR estimator* (10) *satisfies*

$$\hat{A}_m^{PCR} = \widetilde{\Phi}_Y K_{\tilde{Y},\tilde{Y}}^\dagger K_{\tilde{Y},Y} K_{X,\tilde{X}} [\![K_{\tilde{X},\tilde{X}}^\dagger K_{\tilde{X},X} K_{X,\tilde{X}}]\!]_r \widetilde{\Phi}_X^* \tag{11}$$

*requiring $O(2m^3 + 2m^2 n)$ operations, i.e. optimal rates can again be obtained at a cost of at most $O(n^2)$ operations.*

Note that with $m = n$, $\hat{A}_m^{\mathrm{PCR}}$ is equivalent to the kernel DMD estimator [67], also known as kernel analog forecasting (KAF) [4]. The sketched estimator of Proposition 3.2 was also recently derived in [6], albeit without providing theoretical guarantees.

**Reduced Rank Regression (RRR)**    Another way to promote low-rank estimators is to add an explicit rank constraint when minimizing the empirical risk. Combining such a constraint with Tikhonov regularization corresponds to the reduced rank regression [24, 29] estimator:

$$A_\lambda^{\mathrm{RRR}} = \underset{A \in \mathrm{HS} : \mathrm{rk}(A) \leq r}{\arg\min} \hat{\mathcal{R}}_{\mathrm{HS}}(A) + \lambda \|A\|_{\mathrm{HS}}^2. \tag{12}$$

Minimizing Eq. (12) requires solving a $n \times n$ generalized eigenvalue problem. The following proposition introduces the sketched version of this estimator, along with a procedure to compute it which instead requires the solution of a $m \times m$ eigenvalue problem. For $m \asymp \sqrt{n}$, which is enough to guarantee optimal learning rates with minimal assumptions (see Section 4), this represents a reduction from $O(n^3)$ to $O(n\sqrt{n})$ time.

**Proposition 3.3 (Nyström RRR):** *The Nyström RRR estimator can be written as*

$$\hat{A}_{m,\lambda}^{RRR} = [\![P_Y \hat{C}_{YX} P_X (P_X \hat{C} P_X + \lambda I)^{-1/2}]\!]_r (P_X \hat{C} P_X + \lambda I)^{-1/2}. \tag{13}$$

*To compute it, solve the $m \times m$ eigenvalue problem*

$$(K_{\tilde{X},X} K_{X,\tilde{X}} + n\lambda K_{\tilde{X},\tilde{X}})^\dagger K_{\tilde{X},X} K_{Y,\tilde{Y}} K_{\tilde{Y},\tilde{Y}}^\dagger K_{\tilde{Y},Y} K_{X,\tilde{X}} w_i = \sigma_i^2 w_i$$

*for the first $r$ eigenvectors $W_r = [w_1, \dots, w_r]$, appropriately normalized. Then denoting $D_r := K_{\tilde{Y},\tilde{Y}}^\dagger K_{\tilde{Y},Y} K_{X,\tilde{X}} W_r$ and $E_r := (K_{\tilde{X},X} K_{X,\tilde{X}} + n\lambda K_{\tilde{X},\tilde{X}})^\dagger K_{\tilde{X},X} K_{Y,\tilde{Y}} D_r$ it holds*

$$\hat{A}_{m,\lambda}^{RRR} = \widetilde{\Phi}_Y D_r E_r^* \widetilde{\Phi}_X^*. \tag{14}$$

# 4 Learning bounds in operator norm for the sketched estimators

In this section, we state the main theoretical results showing that optimal rates for operator learning with KRR, PCR and RRR can be reached with Nyström estimators.

**Assumptions** We first make two assumptions on the space $\mathcal{H}$ used for the approximation, via its reproducing kernel $k$.

**Assumption 4.1 (Bounded kernel):** *There exists $K < \infty$ such that* $\operatorname{ess\,sup}_{\boldsymbol{x}\sim\pi}\|\phi(\boldsymbol{x})\| \leq K$.

Assumption 4.1 ensures that $\mathcal{H}$ is compactly embedded in $L_\pi^2$ [61, Lemma 2.3], and we denote $\Phi_X^* : \mathcal{H} \to L_\pi^2$ the embedding operator which maps any function in $\mathcal{H}$ to its equivalence class $\pi$-almost everywhere in $L_\pi^2$.

**Assumption 4.2 (Universal kernel):** *The kernel $k$ is universal, i.e.* $\operatorname{cl}(\operatorname{ran}(\Phi_X^*)) = L_\pi^2$.

We refer the reader to [60, Definition 4.52] for a definition of a universal kernel. The third assumption on the RKHS is related to the embedding property from Fischer and Steinwart [18], connected to the embedding of interpolation spaces. For a detailed discussion see Section A.3.

**Assumption 4.3 (Embedding property):** *There exists $\tau \in\ ]0,1]$ and $c_\tau > 0$ such that* $\operatorname{ess\,sup}_{\boldsymbol{x}\sim\pi}\|C_\lambda^{-1/2}\phi(\boldsymbol{x})\|^2 \leq c_\tau\lambda^{-\tau}$.

Next, we make an assumption on the decay of the spectrum of the covariance operator that is of paramount importance for derivation of optimal learning bounds. In the following, $\lambda_i(A)$ and $\sigma_i(A)$ always denote the eigenvalues and singular values of an operator $A$ (in decreasing order).

**Assumption 4.4 (Spectral decay):** *There exists $\beta \in\ ]0,\tau]$ and $c > 0$ such that $\lambda_i(C) \leq ci^{-1/\beta}$.*

This assumption is common in the literature, and we will see that the optimal learning rates depend on $\beta$. It implies the bound $d_{\mathrm{eff}}(\lambda) := \operatorname{tr}(C_\lambda^{-1}C) \lesssim \lambda^{-\beta}$ on the effective dimension, which is a key quantity in the analysis (both statements are actually equivalent, see Section E.2). Note that $d_{\mathrm{eff}}(\lambda) = \mathbf{E}_{\boldsymbol{x}\sim\pi}\|C_\lambda^{-1/2}\phi(\boldsymbol{x})\| \leq \operatorname{ess\,sup}_{\boldsymbol{x}\sim\pi}\|C_\lambda^{-1/2}\phi(\boldsymbol{x})\|$, and thus it necessarily holds $\beta \leq \tau$. For a Gaussian kernel, both $\beta$ and $\tau$ can be chosen arbitrarily close to zero.

Finally, we make an assumption about the regularity of the problem itself. A common assumption occurring in the literature is that $\mathbf{E}[f(X_1)\,|\,X_0 = \cdot] \in \mathcal{H}$ for every $f \in \mathcal{H}$, meaning that one can define the Koopman operator directly on the space $\mathcal{H}$, i.e. the learning problem is *well-specified*. However, this assumption is often too strong. Following [30, D.1] we make a different assumption on the cross-covariance remarking that, irrespectively of the choice of RKHS, it holds true whenever the Koopman operator is self-adjoint (i.e. the dynamics is time-reversible).

**Assumption 4.5 (Regularity of $\mathcal{K}_\pi$):** *There exists $a > 0$ such that $C_{XY}C_{XY}^* \preccurlyeq a^2C^2$.*

**Rates** The risk can be decomposed as $\mathcal{R}_{\mathrm{HS}}(A) = \mathcal{E}_{\mathrm{HS}}(A) + \mathcal{R}_{\mathrm{HS},0}$ where $\mathcal{R}_{\mathrm{HS},0}$ is a constant and $\mathcal{E}_{\mathrm{HS}}(A) := \|\mathcal{K}_\pi\Phi_X^* - \Phi_X^*A^*\|_{\mathrm{HS}}^2$ corresponds to the excess risk (more details in Section B). Optimal learning bounds for the KRR estimator in the context of CME (i.e. in Hilbert-Schmidt norm) have been developed in [33] under Assumptions 4.1 to 4.4 in well-specified and misspecified settings. On the other hand, in the context of dynamical systems, Kostic et al. [29, Theorem 1] report the importance of *reduced rank estimators* that have a small excess risk in operator norm

$$\mathcal{E}(A) := \|\mathcal{K}_\pi\Phi_X^* - \Phi_X^*A^*\|_{\mathcal{H}\to L_\pi^2}^2. \tag{15}$$

The rationale behind considering the operator norm is that it allows to control the error of the eigenvalues approximation and thus of the KMD (3), (4) as discussed below. Optimal learning bounds in operator norm for KRR, PCR and RRR are established in [30]. In this work we show that the same optimal rates remain valid for the *Nyström* KRR, PCR and RRR estimators. According to [29] and [30] these operator norm bounds lead to reliable approximation of the Koompan mode decomposition of Eq. (4).

We now provide our main result.

**Theorem 4.6 (Operator norm error for KRR, i.i.d. data):** *Let assumptions 4.1 to 4.5 hold. Let $(\boldsymbol{x}_i, \boldsymbol{y}_i)_{1\leq i\leq n}$ be i.i.d. samples, and let $P_Y = P_X$ be the projection induced by $m$ Nyström landmarks drawn uniformly from $(\boldsymbol{x}_i)_{1\leq i\leq n}$ without replacement. Let $\lambda = c_\lambda n^{-1/(1+\beta)}$ where $c_\lambda$ is a constant*

*given in the proof, and assume $n \geq (c_\lambda / K^2)^{1+\beta}$. Then it holds with probability at least $1 - \delta$*

$$\mathcal{E}(\hat{A}_{m,\lambda}^{KRR})^{1/2} \lesssim n^{-\frac{1}{2(1+\beta)}} \qquad provided \qquad m \gtrsim \max(1, n^{\tau/(1+\beta)}) \log(n/\delta).$$

The proof is provided in Section E.2, but essentially relies on a decomposition involving the terms $\|C_\lambda^{-1/2}(C_{YX} - \hat{C}_{YX})\|$, $\|C_\lambda^{-1/2}(C - \hat{C})\|$, $\|C_\lambda^{-1/2}(C - \hat{C})C_\lambda^{-1/2}\|$, as well as bounding the quantity $\|P_X^\perp C^{1/2}\|$ where $P_X^\perp$ denotes the projection on the orthogonal of $\mathrm{ran}(P_X)$. All these terms are bounded using two variants of the Bernstein inequality. Note that our results can easily be extended to leverage score sampling of the landmarks by bounding term $\|P_X^\perp C^{1/2}\|$ by means of [53, Lemma 7]; the same rate could then be obtained using a smaller number $m$ of Nyström points.

The rate $n^{-1/(2(1+\beta))}$ is known to be optimal (up to the log factor) in this setting by assuming an additional lower bound on the decay of the covariance's eigenvalues of the kind $\lambda_i(C) \gtrsim i^{-1/\beta}$, see [30, Theorem 7 in D.4]. One can see that without particular assumptions ($\beta = \tau = 1$), we only need the number $m$ of inducing points to be of the order of $\Omega(\sqrt{n})$ in order to get an optimal rates. For $\tau$ fixed, this number increases when $\beta$ decreases (faster decay of the covariance's spectrum), however note that the optimal rate depends on $\beta$ and also improves in this case. The dependence in $\tau$ is particularly interesting, as for instance with a Gaussian kernel it is known that $\tau$ can be chosen arbitrarily closed to zero [33, 18]. In that case, the number $m$ of inducing points can be taken on the order of $\Omega(\log n)$.

Note that a bound for the Nyström KRR estimator has been derived in Hilbert-Schmidt norm by Ahmad et al. [3]. Using the operator norm however allows to derive bounds on the eigenvalues (see discussion below), which is of paramount importance for practical applications. Moreover, we now provide a bound on the error of PCR and RRR estimators, which are not covered in [3].

**Lemma 4.7 (Operator norm error for PCR and RRR, i.i.d. data):** *Under the assumptions of Theorem 4.6, taking $\lambda = c_\lambda n^{-1/(1+\beta)}$ with $c_\lambda$ as in Theorem 4.6, $n \geq (c_\lambda / K^2)^{1+\beta}$, and provided*

$$m \gtrsim \max(1, n^{\tau/(1+\beta)}) \log(n/\delta),$$

*it holds with probability at least $1 - \delta$*

$$\mathcal{E}(\hat{A}_{m,\lambda}^{RRR})^{1/2} \lesssim c_{\mathrm{RRR}} \, n^{-\frac{1}{2(1+\beta)}}, \; \textit{for } r \textit{ s.t. } \sigma_{r+1}(\Phi_X \mathcal{K}_\pi^*) < \min(\sigma_r(\Phi_X \mathcal{K}_\pi^*), n^{-\frac{1}{2(1+\beta)}})$$

$$\textit{and} \quad \mathcal{E}(\hat{A}_m^{PCR})^{1/2} \lesssim c_{\mathrm{PCR}} \, n^{-\frac{1}{2(1+\beta)}}, \; \textit{for } r > n^{\frac{1}{\beta(1+\beta)}},$$

*where $c_{\mathrm{RRR}} = (\sigma_r^2(\Phi_X \mathcal{K}_\pi^*) - \sigma_{r+1}^2(\Phi_X \mathcal{K}_\pi^*))^{-1}$ and $c_{\mathrm{PCR}} = (\sigma_r(\Phi_X) - \sigma_{r+1}(\Phi_X))^{-1}$ are the problem dependant constants.*

Note that when rank of $\mathcal{K}_\pi$ is $r$, then there is no restriction on $r$ for the RRR estimator, while for PCR the choice of $r$ depends on the spectral decay property of the kernel. In general, if $r > n^{\frac{1}{\beta(1+\beta)}}$, then $\sigma_{r+1}(\Phi_X \mathcal{K}_\pi^*) \leq \sigma_{r+1}(\Phi_X) \lesssim n^{-1/(2(1+\beta))}$, which implies that RRR estimator can achieve the same rate of PCR but with smaller rank. Again the rate is sharp (up to the log factor) in this setting [30].

**Koopman mode decomposition** According to [29, Theorem 1], working in operator norm allows us to bound the error of our estimators for dynamic mode decomposition, as well as to quantify how close the eigenpairs $(\hat{\lambda}_i, \hat{\varphi}_i)$ of an estimator $\hat{A}^*$ are to being eigenpairs of the Koopman operator. Namely, recalling that for function $\hat{\varphi}_i$, the corresponding candidate for Koopman eigenfunction in $L_\pi^2$ space is $\Phi_X^* \hat{\varphi}_i$, one has $\|\mathcal{K}_\pi(\Phi_X^* \hat{\varphi}_i) - \hat{\lambda}_i(\Phi_X^* \hat{\varphi}_i)\| / \|\Phi_X^* \hat{\varphi}_i\| \leq \mathcal{E}(\hat{A})^{1/2} \|\hat{\varphi}_i\| / \|\Phi_X^* \hat{\varphi}_i\|$. While eigenvalue and eigenfunction learning rates were studied, under additional assumptions, in [30], where the operator norm error rates were determinant, here, in Section 5, we empirically show that the proposed estimators accurately learn the Koopman spectrum. We refer the reader to Section D for the details on computation of eigenvalues, eigenfunctions and KMD of an estimator in practice.

**Dealing with non-i.i.d. data** The previous results hold for i.i.d. data, which is not a very realistic assumption when learning from sampled trajectories. Our results can however easily be extended to $\beta$-mixing processes by considering random variables $Z_i = \sum_{j=1}^k X_{i+j}$ (thus representing portions of the trajectory) sufficiently separated in time to be nearly independent. We now consider a trajectory

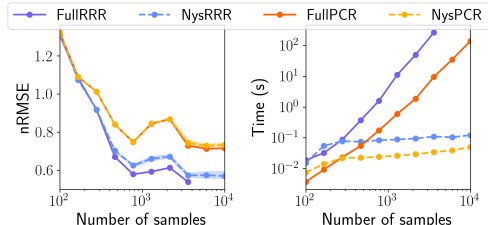
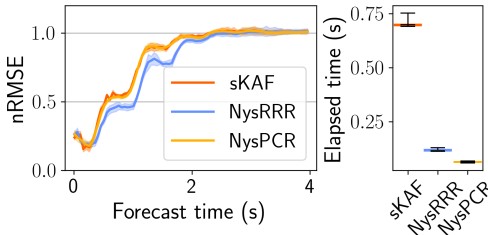

Figure 1: Full and Nyström estimators trained on L63 with increasing $n$. Error (*left*) and running time (*right*) are plotted to show efficiency gains without accuracy loss with the Nyström approximation. RBF($\sigma = 3.5$) kernel, $r = 25$ principal components and $m = 250$ inducing points.

Figure 2: Nyström and sKAF estimators trained on L63 for increasing forecast horizons; the error (*left*) and overall running times (*right*) are shown. We used a RBF kernel with $\sigma = 3.5$, $r = 50$, $m = 250$ (for Nyström methods) and $\sqrt{n} \log n$ random features (for sKAF).

$\boldsymbol{x}_1, \ldots, \boldsymbol{x}_{n+1}$ with $\boldsymbol{x}_1 \sim \pi$ and $\boldsymbol{x}_{t+1} \sim p(\boldsymbol{x}_t, \cdot)$ for $t \in [1, n]$, and use Lemma J.8 (re-stated from [29]) which allows to translate concentration results on the $Z_i$ to concentration on the $X_i$ by means of the $\beta$-mixing coefficients defined as $\beta_X(k) := \sup_{B \in \mathcal{B} \otimes \mathcal{B}} |\rho_k(B) - (\pi \times \pi)(B)|$ where $\rho_k$ denotes the joint probability of $(X_t, X_{t+k})$. Using this result the concentration results provided in appendix can thus be generalized to the $\beta$-mixing setting, and apart from logarithmic dependencies we essentially obtain similar results to the i.i.d. setting except that the sample size $n$ is replaced by $p \approx n/(2k)$.

## 5 Experimental validation

In this section we show how the estimators proposed in section 3 perform in various scenarios, ranging from synthetic low dimensional ODEs to large-scale molecular dynamics simulations. The code for reproducing all experiments is available online. Our initial aim is to demonstrate the speed of NysPCR and NysRRR, compared to the recently proposed alternative Streaming KAF (sKAF) [20]. Then we show that their favorable scaling properties make it possible to train on large molecular dynamics datasets without any subsampling. In particular we run a metastability analysis of the alanine dipeptide and the Trp-cage protein, showcasing the accuracy of our models' eigenvalue and eigenfunction estimates, as well as their efficiency on massive datasets ($> 500\,000$ points)

**Efficiency Benchmarks on Lorenz '63** The chaotic Lorenz '63 system [36] consists of 3 ODEs with no measurement noise. With this toy dynamical system we can easily compare the Nyström estimators to two alternatives: 1. the corresponding *exact* estimators and 2. the sKAF algorithm which also uses randomized linear algebra to improve the efficiency of PCR. In this setting we sample long trajectories from the system, keeping the first points for training (the number of training points varies for the first experiment, and is fixed to $10\,000$ for the second, see fig. 2), and the subsequent ones for testing. In fig. 1 we compare the run-time and accuracy with of NysPCR and NysRRR versus their full counterparts. To demonstrate the different scaling regimes we fix the number of inducing points to 250 and increase the number of data points $n$. The accuracy of the two solvers (as measured with the normalized RMSE metric (nRMSE) [20] on the first variable) is identical for PCR and close for RRR, but the running time of the approximate solvers increases much slower with $n$ than that of the exact solvers. Each experiment is repeated 20 times to display error bars over the choice of Nyström centers. In the second experiment, shown in fig. 2, we reproduce the setting of [20] by training at increasingly long forecast horizons. Plotting the nRMSE we verify that sKAF and NysPCR converge to very similar accuracy values, although NysPCR is approximately 10 times faster. NysRRR instead offers slightly better accuracy, at the expense of a higher running time compared to NysPCR. Error bars are the standard deviation of nRMSE over 5 successive test sets with $10\,000$ points each. In section K we show additional experiments including a comparison to the Nyström KRR estimator.

**Molecular dynamics datasets** An important application of Koopman operator theory is in the analysis of molecular dynamics (MD) datasets, where the evolution of a molecule's atomic positions as they evolve over time is modelled. Interesting systems are very high dimensional, with hundreds or thousands of atoms. Furthermore, trajectories are generated at very short time intervals ($< 1\,\text{ns}$) but

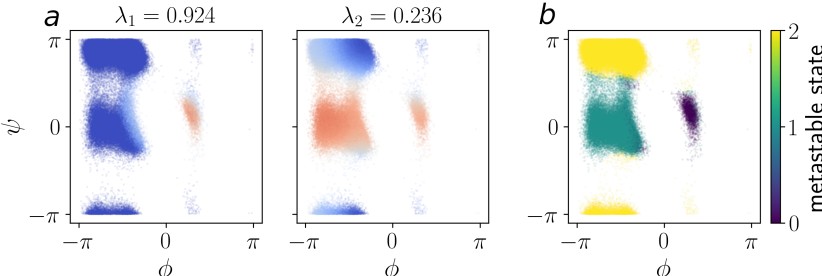

Figure 3: Dynamics of the alanine dipeptide (lag-time 100), Nyström RRR model. On the left the Ramachandran plot of all protein conformations is colored with the value of the first two non-constant eigenfunctions evaluated on the 45-d space. On the right color indicates the discrete state obtained by clustering the dynamics projected onto the first eigenfunctions (using PCCA+ with 3 states). The obtained clusters match the main areas of the Ramachandran plot.

interesting events (e.g. protein folding/unfolding) occur at timescales on the order of at least $10\,\mu s$, so that huge datasets are needed to have a few samples of the rare events. The top eigenfunctions of the Koopman operator learned on such trajectories can be used to project the high-dimensional state space onto low-dimensional coordinates which capture the long term, slow dynamics.

We take three $250\,ns$ long simulations sampled at $1\,ps$ of the alanine dipeptide [64], which is often taken as a model system for molecular dynamics [47, 46]. We use the pairwise distances between heavy atoms as features, yielding a 45-dimensional space. We train a NysRRR model with $10\,000$ centers on top of the full dataset ($449\,940$ points are used for training, the rest for validation and testing) with lag time $100\,ps$, and recover a 2-dimensional representation which correlates well with the $\phi, \psi$ backbone dihedral angles of the molecule, known to capture all relevant long-term dynamics. Figure 3a shows the top two eigenfunctions overlaid onto $\phi, \psi$, the first separates the slowest transition between low and high $\phi$; the second separates low and high $\psi$. The implied time-scales from the first two non-trivial eigenvalues are $1262\,ps$ and $69\,ps$, which are close to the values reported by Nüske et al. [47] ($1400\,ps$ and $70\,ps$) who used a more complex post-processing procedure to identify time-scales. We then train a PCCA+ [16] model on the first three eigenfunctions to obtain three states, as shown in fig. 3b. PCCA+ acts on top of a fine clustering (in our case obtained with k-means, $k = 50$), to find the set of maximally stable states by analyzing transitions between the fine clusters. The coarse clusters clearly correspond to the two transitions described above.

Finally we take a $208\,\mu s$ long simulation of the fast-folding Trp-cage protein [35], sampled every $0.2\,ns$. Again, the states are the pairwise distances between non-hydrogen atoms belonging to the protein, in $10\,296$ dimensions. A NysRRR model is trained on $626\,370$ points, using $5000$ centers in approximately 10 minutes. Note that without sketching this would be a completely intractable problem. Using a lag-time of $10\,ns$ we observe a spectral gap between the third and fourth eigenvalues, hence we train a PCCA+ model on the first 3 eigenfunctions to obtain the states shown in fig. 4. The first non-trivial Koopman eigenvector effectively distinguishes between the folded (state 1) and unfolded states as is evident from the first row of fig. 4. The second one instead can be used to identify a partially folded state of the protein (state 0), as can be seen from the insets in fig. 4.

## 6 Conclusions

We introduced three efficient kernel-based estimators of the Koopman operator relying on random projections, and provided a bound on their excess risk in operator norm – which is of paramount importance to control the accuracy of Koopman mode decomposition. Random projections allow to process efficiently even the longest trajectories, and these gains come for free as our estimators still enjoy optimal theoretical learning rates. We leave for future work the refinement our analysis under e.g. an additional source condition assumption or in the misspecified setting. Another future research direction shall be to devise ways to further reduce the computational complexity of the estimators.

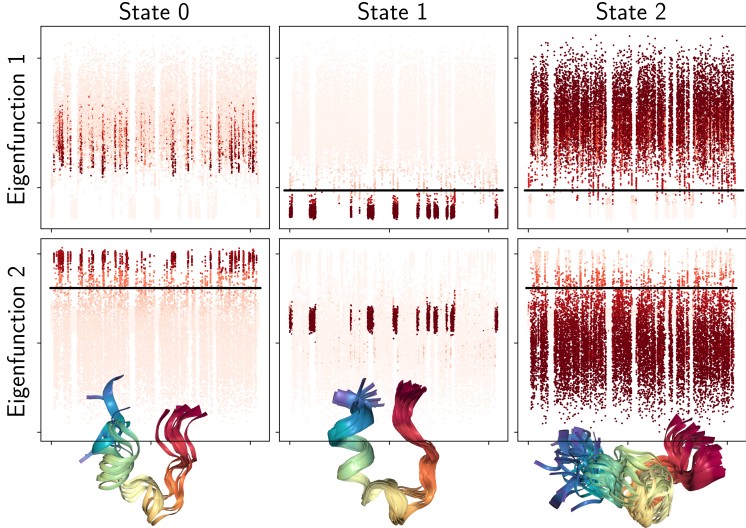

Figure 4: First eigenfunctions for Trp-cage dynamics, colored according to the membership probability for each state in a PCCA+ model. The bottom insets show a few overlaid structures from each state. The first eigenfunction exhibits a strong linear separation between state 1 (folded) and the other states, which is highlighted by the black lines. The second separates between state 0 (partially folded) ant the rest. NysRRR model trained with $m = 5000$, $r = 10$, RBF($\sigma = 0.02$) kernel, $\lambda = 10^{-10}$.

## 7    Acknowledgements

This paper is part of a project that has received funding from the European Research Council (ERC) under the European Union's Horizon 2020 research and innovation programme (grant agreement No. 819789). L. R. acknowledges the financial support of the European Research Council (grant SLING 819789), the AFOSR projects FA9550-18-1-7009, FA9550-17-1-0390 and BAA-AFRL-AFOSR-2016-0007 (European Office of Aerospace Research and Development), the EU H2020-MSCA-RISE project NoMADS - DLV-777826, and the Center for Brains, Minds and Machines (CBMM), funded by NSF STC award CCF-1231216. M. P., V. K. and P. N. acknowledge financial support from PNRR MUR project PE0000013-FAIR and the European Union (Projects 951847 and 101070617).

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

# Appendices for: "Estimating Koopman operators with sketching to provably learn large scale dynamical systems"

## A   Setting and notations

### A.1   Operators and notations

We define the following operators:

- $\Phi_X : L^2_\pi \to \mathcal{H}$, defined by $\Phi_X f = \int_\mathcal{X} f(x)\phi(x)\,\mathrm{d}\pi(x)$ for any $f \in L^2_\pi$.
- $\Phi_X^* : \mathcal{H} \to L^2_\pi$, defined by $\Phi_X^* h = \langle h, \phi(\cdot)\rangle_\mathcal{H}$ for any $h \in \mathcal{H}$ (i.e. the embedding operator mapping a function to its $\pi$-equivalence class in $L^2_\pi$).
- $\Phi_{Y|X} : L^2_\pi \to \mathcal{H}$, defined by $\Phi_{Y|X} = \Phi_X \mathcal{K}_\pi^*$.
- $\Phi_{Y|X}^* : \mathcal{H} \to L^2_\pi$, defined by $\Phi_{Y|X}^* = \mathcal{K}_\pi \Phi_X^*$.
- $C : \mathcal{H} \to \mathcal{H}$ defined as $C = \mathbf{E}_{x\sim\pi}\phi(x)\otimes\phi(x) = \Phi_X\Phi_X^*$, satisfying $\mathrm{tr}(C) \le K^2$. Note that under our assumptions, this also corresponds to the covariance of $Y$.
- $C_{XY} := \mathbf{E}_{(x,y)\sim\rho}\phi(x)\otimes\phi(y) = \Phi_X\Phi_{Y|X}^*$.

As well as the following discretized variants:

- $\hat{\Phi}_X : \mathbb{R}^n \to \mathcal{H}$, defined by $\hat{\Phi}_X v = \sum_{i=1}^n v_i\phi(x_i)$ for any $v = [v_1,\dots,v_n] \in \mathbb{R}^n$
- $\hat{\Phi}_X^* : \mathcal{H} \to \mathbb{R}^n$, defined by $\hat{\Phi}_X^* h = [\langle\phi(x_1),h\rangle_\mathcal{H},\dots,\langle\phi(x_n),h\rangle_\mathcal{H}]^T$ for any $h \in \mathcal{H}$
- $\hat{\Phi}_{Y|X} : \mathbb{R}^n \to \mathcal{H}$, defined by $\hat{\Phi}_{Y|X} v = \sum_{i=1}^n v_i\phi(y_i)$ for any $v = [v_1,\dots,v_n] \in \mathbb{R}^n$.
- $\hat{\Phi}_{Y|X}^* : \mathcal{H} \to \mathbb{R}^n$, defined by $\hat{\Phi}_{Y|X}^* h = [\langle\phi(y_1),h\rangle_\mathcal{H},\dots,\langle\phi(y_n),h\rangle_\mathcal{H}]^T$ for any $h \in \mathcal{H}$.
- $\hat{C} = \frac{1}{n}\hat{\Phi}_X\hat{\Phi}_X^* = \frac{1}{n}\sum_{i=1}^n \phi(x_i)\otimes\phi(x_i) \in \mathcal{L}(\mathcal{H})$ is the empirical covariance.

The Nyström discretized operators are obtained by applying the kernel map to $m \ll n$ inducing points $\{\tilde{x}_j\}_{j=1}^m \subset \{x_j\}_{j=1}^n$ and $\{\tilde{y}_j\}_{j=1}^m \subset \{y_j\}_{j=1}^n$:

- $\widetilde{\Phi}_X : \mathbb{R}^m \to \mathcal{H}$ such that $\widetilde{\Phi}_X w = \sum_{j=1}^m w_j\phi(\tilde{x}_j)$.
- $\widetilde{\Phi}_Y : \mathbb{R}^m \to \mathcal{H}$ such that $\widetilde{\Phi}_Y w = \sum_{j=1}^m w_j\phi(\tilde{y}_j)$.

Furthermore denote by $P_X$ and $P_Y$ the orthogonal projections onto $\mathrm{span}\,\widetilde{\Phi}_X$ and $\mathrm{span}\,\widetilde{\Phi}_Y$ respectively.

One important quantity to derive the rates is the so-called effective dimension, defined as

$$d_{\mathrm{eff}}(\lambda) := \mathrm{tr}(C_\lambda^{-1}C).$$

where $C_\lambda := C + \lambda I$.

### A.2   Conditional mean embedding

For any $x \in \mathcal{X}$, we denote $\mu_p(x)$ the conditional mean embedding associated to the transition kernel defined as

$$\mu_p(x) := \mathbf{E}\big[\phi(X_{t+1})|X_t = x\big] = \int \phi(y)p(x,\mathrm{d}y)$$

The following lemma provides a characterization of $\Phi_{Y|X}^*$ in terms of the conditional mean embedding.

**Lemma A.1:** *We have the following relations:*

$$\Phi_{Y|X}f = \int_{\mathcal{X}} f(x)\mu_p(x)\,\mathrm{d}\pi(x), \quad f \in L^2_\pi \tag{16}$$

$$(\Phi^*_{Y|X}f)(x) = \langle f, \mu_p(x)\rangle, \quad f \in \mathcal{H} \tag{17}$$

$$\Phi_{Y|X}\Phi^*_{Y|X} = \mathbf{E}_{x\sim\pi}\mu_p(x) \otimes \mu_p(x) \tag{18}$$

*Proof of Lemma A.1:* For the first property:

$$(\Phi^*_{Y|X}f)(x) = (\mathcal{K}_\pi(\Phi^*_X f))(x) \tag{19}$$

$$= \int (\Phi^*_X f)(y)p(x,\mathrm{d}y) \tag{20}$$

$$= \int f(y)p(x,\mathrm{d}y) \tag{21}$$

$$= \langle f, \int \phi(y)p(x,\mathrm{d}y)\rangle = \langle f, \mu_p(x)\rangle \tag{22}$$

where we used that $f$ and $\Phi^*_X f$ coincide $\pi$-almost everywhere. The second property is a direct consequence of the definition of the adjoint. For (18), we simply use (17) and the definition of $\Phi_{Y|X}$ to get

$$\Phi_{Y|X}(\Phi^*_{Y|X}f) = \int \langle f, \mu_p(z)\rangle\mu_p(z)\,\mathrm{d}\pi(z) = \left(\int \mu_p(z)\mu_p(z)^*\,\mathrm{d}\pi(z)\right)f.$$

$\square$

## A.3  Power spaces

We now define the $\alpha$-power space $[\mathcal{H}]^\alpha_\pi$ in order to provide some intuition regarding Assumption 4.3.

By Assumption 4.1, $\mathrm{tr}(C) = \int \mathrm{tr}(\phi(x)\otimes\phi(x))\,\mathrm{d}\pi(x) \leq K^2$ and thus $C$ is trace-class (and compact). By [18], there exists a non-increasing summable sequence $(\mu_i)_{i\in I}$ for an at most countable index set $I$, a family $(e_i)_{i\in I} \in \mathcal{H}$ s.t. $(\Phi^*_X e_i)_{i\in I}$ is an orthonormal basis of $\overline{\mathrm{span}\,\Phi^*_X} \subseteq L^2_\pi$ and $(\mu_i^{1/2}e_i)_{i\in I}$ is an orthonormal basis of $(\ker\Phi^*_X)^\perp \subseteq \mathcal{H}$ such that

$$C = \sum_{i\in I} \mu_i\langle\cdot, \mu_i^{1/2}e_i\rangle_{\mathcal{H}}\mu_i^{1/2}e_i.$$

For $\alpha \geq 0$, we now define the $\alpha$-power space as

$$[\mathcal{H}]^\alpha_\pi := \left\{ \sum_{i\in I} a_i\mu_i^{\alpha/2}\Phi^*_X e_i \;\middle|\; (a_i)_{i\in I} \in \ell_2(I) \right\} \subseteq L^2_\pi$$

equipped with norm

$$\left\|\sum_{i\in I} a_i\mu_i^{\alpha/2}\Phi^*_X e_i\right\|_{[\mathcal{H}]^\alpha_\pi} := \|(a_i)_{i\in I}\|_{\ell_2(I)}.$$

We can now make the following assumption regarding the embedding of the power spaces into $L^\infty_\pi$.

**Assumption A.2 (Embedding):** *There exists $\tau \in [\beta, 1]$ such that $c_\tau := \|[\mathcal{H}]^\tau_\pi \hookrightarrow L^\infty_\pi\|^2 < \infty$.*

We stress that Assumption A.2 implies in particular Assumption 4.3, and is a common assumption in the literature, see for instance [18].

## B  Expression of the risk

We have the following risk decomposition.

**Lemma B.1:** *The risk can alternatively be written*

$$\mathcal{R}_{\mathrm{HS}}(A) = \mathbf{E}_{(x,y)\sim\rho}\|\phi(y) - A\phi(x)\|^2$$
$$= \mathcal{R}_{\mathrm{HS},0} + \mathcal{E}_{\mathrm{HS}}(A)$$
$$\textit{where}\quad \mathcal{R}_{\mathrm{HS},0} := \|\Phi_X\|_{\mathrm{HS}}^2 - \|\Phi_{Y|X}\|_{\mathrm{HS}}^2$$
$$= \int \|\mu_p(x) - \phi(y)\|^2 \, \mathrm{d}\rho(x,y)$$
$$\textit{and}\quad \mathcal{E}_{\mathrm{HS}}(A) := \|\Phi_{Y|X} - A\Phi_X\|_{\mathrm{HS}}^2$$
$$= \int \|\mu_p(x) - A\phi(x)\|^2 \, \mathrm{d}\pi(x).$$

*where* $\inf_{A\in\mathrm{HS}(\mathcal{H})} \mathcal{E}_{\mathrm{HS}}(A) = 0$, *and thus we interpret* $\mathcal{E}_{\mathrm{HS}}$ *as the excess risk.*

*Proof of Lemma B.1:*  Let $(h_i)_{i\in\mathbb{N}}$ be an orthonormal basis of $\mathcal{H}$. Then

$$\mathcal{E}_{\mathrm{HS}}(A) := \|\Phi_{Y|X} - A\Phi_X\|_{\mathrm{HS}}^2$$
$$= \sum_{i\in\mathbb{N}} \|\Phi_{Y|X}^* h_i - \Phi_X^* A^* h_i\|_{L_\pi^2}^2$$
$$= \sum_{i\in\mathbb{N}} \int \left((\Phi_{Y|X}^* h_i)(x) - \langle A^* h_i, \phi(x)\rangle_\mathcal{H}\right)^2 \mathrm{d}\pi(x)$$
$$\text{(by (17))}\quad = \sum_{i\in\mathbb{N}} \int \left(\langle h_i, \mu_p(x)\rangle_\mathcal{H} - \langle h_i, A\phi(x)\rangle_\mathcal{H}\right)^2 \mathrm{d}\pi(x)$$
$$= \int \|\mu_p(x) - A\phi(x)\|^2 \, \mathrm{d}\pi(x).$$

It holds

$$\mathcal{R}_{\mathrm{HS},0} = \int \|\mu_p(x) - \phi(y)\|^2 \, \mathrm{d}\rho(x,y)$$
$$= \int \left(\|\mu_p(x)\|^2 - 2\langle\mu_p(x), \phi(y)\rangle_\mathcal{H} + \mathrm{tr}\,\phi(y)\phi(y)^*\right) \mathrm{d}\rho(x,y)$$
$$= \int \|\mu_p(x)\|^2 \, \mathrm{d}\pi(x) - 2\int \left\langle\mu_p(x), \int \phi(y)p(x,\mathrm{d}y)\right\rangle_\mathcal{H} \mathrm{d}\pi(x) + \int\int \mathrm{tr}\,\phi(y)\phi(y)^* p(x,\mathrm{d}y)\,\mathrm{d}\pi(x)$$
$$\overset{(i)}{=} -\int \|\mu_p(x)\|^2 \, \mathrm{d}\pi(x) + \int \mathrm{tr}\big(\phi(y)\phi(y)^*\big) \, \mathrm{d}\pi(y)$$
$$= -\mathrm{tr}\left(\int \mu_p(x)\mu_p(x)^* \, \mathrm{d}\pi(x)\right) + \mathrm{tr}(C)$$
$$= -\mathrm{tr}\left(\Phi_{Y|X}\Phi_{Y|X}^*\right) + \mathrm{tr}(\Phi_X\Phi_X^*)$$

where we used the invariance property of $\pi$ in $(i)$ and Lemma A.1 for the last inequality. Then one can easily check that the sum of both corresponds to the full risk defined in (5):

$$
\begin{aligned}
\mathcal{R}_{\mathsf{HS},0} + \mathcal{E}_{\mathsf{HS}}(A) &= \int \|\mu_p(x) - \phi(y)\|^2 \, \mathrm{d}\rho(x,y) + \int \|\mu_p(x) - A\phi(x)\|^2 \, \mathrm{d}\pi(x) \\
&= \int \left( \|\mu_p(x)\|^2 - 2\langle \mu_p(x), \int \phi(y)p(x,\mathrm{d}y)\rangle + \int \|\phi(y)\|^2 p(x,\mathrm{d}y) \right) \mathrm{d}\pi(x) \\
&\quad + \int \left( \|\mu_p(x)\|^2 - 2\langle \mu_p(x), A\phi(x)\rangle + \|A\phi(x)\|^2 \right) \mathrm{d}\pi(x) \\
&= \int \left( \int \|\phi(y)\|^2 p(x,\mathrm{d}y) - 2\langle \int \phi(y)p(x,\mathrm{d}y), A\phi(x)\rangle + \|A\phi(x)\|^2 \right) \mathrm{d}\pi(x) \\
&= \int \left( \|\phi(y)\|^2 - 2\langle \phi(y), A\phi(x)\rangle + \|A\phi(x)\|^2 \right) \mathrm{d}\rho(x,y) \\
&= \int \|\phi(y) - A\phi(x)\|^2 \, \mathrm{d}\rho(x,y) = \mathcal{R}_{\mathsf{HS}}(A).
\end{aligned}
$$

$\square$

## C   Expression of the estimators

In this section we give proofs of propositions 3.1 to 3.3 on how to efficiently compute the Nyström estimators.

For all three – KRR, PCR and RRR – estimators, the starting point is their respective *full* estimator which can be derived by following the first-order optimality criterion for the following minimization problems

$$
\textbf{Full KRR:} \qquad \hat{A}_\lambda^{\mathsf{KRR}} = \underset{A \in \mathcal{H} \to \mathcal{H}}{\arg\min} \|\hat{\Phi}_{Y|X} - A\hat{\Phi}_X\|_{\mathsf{HS}}^2 + \lambda\|A\|_{\mathsf{HS}}^2 \tag{23}
$$

$$
\textbf{Full PCR:} \qquad \hat{A}^{\mathsf{PCR}} = \underset{A \in \mathcal{H} \to \mathcal{H}}{\arg\min} \|\hat{\Phi}_{Y|X} - A\Pi_r\hat{\Phi}_X\|_{\mathsf{HS}}^2 \tag{24}
$$

$$
\textbf{Full RRR:} \qquad \hat{A}_\lambda^{\mathsf{RRR}} = \underset{A \in \mathcal{H} \to \mathcal{H}:\mathrm{rk}(A)\leq r}{\arg\min} \|\hat{\Phi}_{Y|X} - A\hat{\Phi}_X\|_{\mathsf{HS}}^2 + \lambda\|A\|_{\mathsf{HS}}^2 \tag{25}
$$

where $\Pi_r$ is the orthogonal projection onto the top-r eigenvectors of $\hat{C}$.

To derive the Nyström estimators, we project the embedded data $\hat{\Phi}_X$, $\hat{\Phi}_{Y|X}$ onto the span of the embedded inducing points – $P_X\hat{\Phi}_X$, $P_Y\hat{\Phi}_{Y|X}$ – and then express the resulting estimators as $\widetilde{\Phi}_Y W \widetilde{\Phi}_X^*$ with $W \in \mathbb{R}^{m \times m}$. This form is particularly useful for later computing forecasts, eigenfunctions and Koopman modes with the estimator. In particular the following equalities for the projection (shown here for $P_X$ but equivalently exist for $P_Y$)

$$
P_X = P_X P_X = \widetilde{\Phi}_X (\widetilde{\Phi}_X^* \widetilde{\Phi}_X)^\dagger \widetilde{\Phi}_X^* = \widetilde{\Phi}_X^{*\dagger} \widetilde{\Phi}_X^* = \widetilde{\Phi}_X \widetilde{\Phi}_X^\dagger,
$$

and the characterization of $P_X$ through the SVD of $\widetilde{\Phi}_X = U\Sigma V^*$, such that $P_X = UU^*$.

### C.1   Nyström KRR

We begin with the Nyström KRR estimator, providing an alternative but equivalent description in lemma C.1.

**Lemma C.1 (Expression of the KRR regularization):** *Let $U$ be such that $P_X = UU^*$, $U^*U = I$. Then it holds*

$$
g_{\mathsf{KRR}}(\hat{C}) := P_X(P_X\hat{C}P_X + \lambda I)^{-1} = U(U^*\hat{C}U + \lambda I)^{-1}U^*. \tag{26}
$$

*Proof of Lemma C.1:* Using $U^*U = I$, it holds $(U^*\hat{C}U + \lambda I)U^* = U^*(UU^*\hat{C}UU^* + \lambda I)$ and thus $U^*(UU^*\hat{C}UU^* + \lambda I)^{-1} = (U^*\hat{C}U + \lambda I)^{-1}U^*$. As a consequence,

$$
\begin{aligned}
g_{\text{KRR}}(\hat{C}) &= P_X(P_X\hat{C}P_X + \lambda I)^{-1} \\
&= UU^*(UU^*\hat{C}UU^* + \lambda I)^{-1} \\
&= U(U^*\hat{C}U + \lambda I)^{-1}U^*.
\end{aligned}
$$

$\square$

Then we can provide the computatable formulas for Nyström KRR

**Proposition C.2 (Nyström KRR):** *The Nyström KRR estimator, obtained by projection of eq. (23) is*

$$
\begin{aligned}
\hat{A}_{m,\lambda}^{KRR} &= P_Y\hat{C}_{YX}P_X(P_X\hat{C}P_X + \lambda I)^{-1} \\
&= \widetilde{\Phi}_Y K_{\tilde{Y},\tilde{Y}}^\dagger K_{\tilde{Y},Y} K_{X,\tilde{X}}(K_{\tilde{X},X}K_{X,\tilde{X}} + n\lambda K_{\tilde{X},\tilde{X}})^\dagger \widetilde{\Phi}_X^*.
\end{aligned}
$$

*Proof of Proposition C.2:* Using the definition in eq. (26), and lemma C.1, we have

$$
\begin{aligned}
\hat{A}_{m,\lambda}^{\text{KRR}} &= P_Y\hat{C}_{YX}g_{\text{KRR}}(\hat{C}) \\
&= P_Y\hat{C}_{YX}U(U^*\hat{C}U + \lambda I)^{-1}U^* \\
&= P_Y\hat{C}_{YX}U\Sigma V^*V\Sigma^{-1}(U^*\hat{C}U + \lambda I)^{-1}\Sigma^{-1}V^*V\Sigma U^*
\end{aligned}
$$

Now using the fact that $\Sigma, V, V^*$ and $U^*\hat{C}U + \lambda I$ are full-rank, it holds [2, eq. (20)]

$$
\begin{aligned}
\hat{A}_{m,\lambda}^{\text{KRR}} &= P_Y\hat{C}_{YX}\widetilde{\Phi}_X(V^*)^\dagger(\Sigma U^*\hat{C}U\Sigma + \lambda\Sigma^2)^\dagger V^\dagger\widetilde{\Phi}_X^* \\
&= P_Y\hat{C}_{YX}\widetilde{\Phi}_X(V\Sigma U^*\hat{C}U\Sigma V^* + \lambda V\Sigma^2 V^*)^\dagger\widetilde{\Phi}_X^*.
\end{aligned}
$$

Finally, by definition of $P_Y, \hat{C}_{YX}$ and $\hat{C}$,

$$
\begin{aligned}
\hat{A}_{m,\lambda}^{\text{KRR}} &= \widetilde{\Phi}_Y(\widetilde{\Phi}_Y^*\widetilde{\Phi}_Y)^\dagger\widetilde{\Phi}_Y^*\hat{\Phi}_{Y|X}\hat{\Phi}_X^*\widetilde{\Phi}_X(\widetilde{\Phi}_X^*\hat{\Phi}_X\hat{\Phi}_X^*\widetilde{\Phi}_X + n\lambda\widetilde{\Phi}_X^*\widetilde{\Phi}_X)^\dagger\widetilde{\Phi}_X^* \\
&= \widetilde{\Phi}_Y K_{\tilde{Y},\tilde{Y}}^\dagger K_{\tilde{Y},Y} K_{X,\tilde{X}}(K_{X,\tilde{X}}K_{X,\tilde{X}} + n\lambda K_{\tilde{X},\tilde{X}})^\dagger\widetilde{\Phi}_X^*.
\end{aligned}
$$

$\square$

**Remark C.1 (Alternative derivation of the Nyström KRR estimator):** *Note that the Nyström KRR estimator can equivalently be derived as the solution to a variational problem similar to eq. (23), where the operator $A$ is restricted to operate between spaces $\mathcal{H}_{\tilde{X}} := \text{span}\,\widetilde{\Phi}_X$ and $\mathcal{H}_{\tilde{Y}} := \text{span}\,\widetilde{\Phi}_Y$.*

## C.2   Nyström PCR

Define the following filter on the spectrum of $P_X\hat{C}P_X$: $g_{\text{PCR}}(\hat{C}) = [\![P_X\hat{C}P_X]\!]_r^\dagger$, which truncates it to the first $r$ components before taking the pseudo-inverse. The Nyström PCR estimator, obtained by projection of eq. (24) is

$$
\hat{A}_m^{\text{PCR}} = P_Y\hat{C}_{YX}g_{\text{PCR}}(\hat{C}). \tag{27}
$$

The next proposition provides an efficiently implementable version of the PCR estimator.

**Proposition C.3 (Nyström PCR):** *The sketched PCR estimator eq. (27) satisfies*

$$
\hat{A}_m^{\text{PCR}} = \widetilde{\Phi}_Y K_{\tilde{Y},\tilde{Y}}^\dagger K_{\tilde{Y},Y} K_{X,\tilde{X}}[\![K_{\tilde{X},\tilde{X}}^\dagger K_{\tilde{X},X}K_{X,\tilde{X}}]\!]_r\widetilde{\Phi}_X^* \tag{28}
$$

*Proof of Proposition C.3:* We begin by computing the decomposition of $P_X \hat{C} P_X$ which is necessary to obtain $g_{\mathrm{PCR}}(\hat{C})$. The following expressions are equivalent [43, Proposition 3] for determining its eigenvectors $\tilde{h}$ and eigenvalues $\lambda$:

$$UU^*\hat{C}UU^*\tilde{h} = \lambda\tilde{h}$$

$$U^*\hat{C}Uh = \lambda h, \qquad \tilde{h} = Uh.$$

Let the truncated eigenvalues be $\Lambda_r = \mathrm{diag}[\lambda_1, \ldots, \lambda_r]$ and the eigenvectors be $H_r = [h_1, \ldots, h_r]$. Then $\tilde{H}_r = UH_r$ must be normalized such that $\tilde{H}_r^*\tilde{H}_r = H_r^*U^*UH_r = I$. The rank-r truncation $[\![P_X\hat{C}P_X]\!]_r$ is a projection onto $\tilde{H}_r\tilde{H}_r^*$:

$$[\![P_X\hat{C}P_X]\!]_r^\dagger = (UU^*\hat{C}UU^*(UH_r)(UH_r)^*)^\dagger = (UH\Lambda H^*H_rH_r^*U^*)^\dagger = UH_r\Lambda_r^{-1}H_r^*U^*$$

where we used that $U^*\hat{C}U = H\Lambda H^*$.
Now substitute $U = \widetilde{\Phi}_X V\Sigma^{-1}$ to simplify the eigendecomposition of $U^*\hat{C}U$:

$$\Sigma^{-1}V^*K_{\tilde{X},X}K_{X,\tilde{X}}V\Sigma^{-1}h = \lambda h$$

$$V\Sigma^{-2}V^*K_{\tilde{X},X}K_{X,\tilde{X}}d = \lambda d, \qquad h = \Sigma^{-1}V^*K_{\tilde{X},X}K_{X,\tilde{X}}d. \tag{29}$$

where $V\Sigma^{-2}V^* = K_{\tilde{X},\tilde{X}}^\dagger$. Denote by $D_r = [d_1, \ldots, d_r]$ the truncated eigenvectors such that $H_r = \Sigma^{-1}V^*K_{\tilde{X},X}K_{X,\tilde{X}}D_r$, normalized such that $H^*H = D^*K_{\tilde{X},X}K_{X,\tilde{X}}K_{\tilde{X},\tilde{X}}^\dagger K_{\tilde{X},X}K_{X,\tilde{X}}D = I$,

$$\begin{aligned}
UH_r\Lambda_r^{-1}H_r^*U^* &= \widetilde{\Phi}_X K_{\tilde{X},\tilde{X}}^\dagger K_{\tilde{X},X}K_{X,\tilde{X}}D_r\Lambda_r^{-1}D_r^*K_{\tilde{X},X}K_{X,\tilde{X}}K_{\tilde{X},\tilde{X}}^\dagger \widetilde{\Phi}_X^* \\
&= \widetilde{\Phi}_X D_r\Lambda_r D_r^*\widetilde{\Phi}_X^* \\
&= \widetilde{\Phi}_X [\![K_{\tilde{X},\tilde{X}}^\dagger K_{\tilde{X},X}K_{X,\tilde{X}}]\!]_r\widetilde{\Phi}_X^*.
\end{aligned}$$

Finally, we can plug the pieces together to get

$$P_Y\hat{C}_{YX}[\![P_X\hat{C}P_X]\!]_r^\dagger = \widetilde{\Phi}_Y K_{\tilde{Y},\tilde{Y}}^\dagger K_{\tilde{Y},Y}K_{X,\tilde{X}}[\![K_{\tilde{X},\tilde{X}}^\dagger K_{\tilde{X},X}K_{X,\tilde{X}}]\!]_r\widetilde{\Phi}_X^*.$$

$\square$

**Remark C.2 (Variational problem for Nyström PCR):** *Note that, unlike the NysKRR estimator, the variational problem for NysPCR where the operator is restricted to $A : \mathcal{H}_{\widetilde{X}} \to \mathcal{H}_{\widetilde{Y}}$ is not equivalent to the one obtained in proposition C.3 by projecting the covariance operator. In fact, the former does not take the full covariance into account when computing the low-rank projection, but just the Nyström points.*

## C.3 Nyström RRR

The Nyström RRR estimator does not correspond to a specific spectral filter. We can nonetheless compute it starting from the expression of the exact empirical estimator [29], projecting the covariance operators, and rearranging the expression to result in a finite-dimensional procedure.

**Proposition C.4 (Nyström RRR):** *The sketched RRR estimator can be written as*

$$\hat{A}_{m,\lambda}^{RRR} = [\![P_Y\hat{C}_{YX}P_X(P_X\hat{C}P_X + \lambda I)^{-1/2}]\!]_r(P_X\hat{C}P_X + \lambda I)^{-1/2}. \tag{30}$$

*To compute it, solve the $m \times m$ eigenvalue problem*

$$(K_{\tilde{X},X}K_{X,\tilde{X}} + n\lambda K_{\tilde{X},\tilde{X}})^\dagger K_{\tilde{X},X}K_{Y,\tilde{Y}}K_{\tilde{Y},\tilde{Y}}^\dagger K_{\tilde{Y},Y}K_{X,\tilde{X}}w_i = \sigma_i^2 w_i$$

*for the first $r$ eigenvectors $W_r = [w_1, \ldots, w_r]$, normalized such that $W_r^*K_{\tilde{X},X}K_{Y,\tilde{Y}}K_{\tilde{Y},\tilde{Y}}^\dagger K_{\tilde{Y},Y}K_{X,\tilde{X}}W_r = I$. Then let $D_r := K_{\tilde{Y},\tilde{Y}}^\dagger K_{\tilde{Y},Y}K_{X,\tilde{X}}W_r$ and $E_r := (K_{\tilde{X},X}K_{X,\tilde{X}} + n\lambda K_{\tilde{X},\tilde{X}})^\dagger K_{\tilde{X},X}K_{Y,\tilde{Y}}U_r$, such that the following holds*

$$\hat{A}_{m,\lambda}^{RRR} = \widetilde{\Phi}_Y D_r E_r^*\widetilde{\Phi}_X^*. \tag{31}$$

*Proof of Proposition C.4:* Let $B := n^{1/2} P_Y \hat{C}_{YX} P_X (P_X \hat{C} P_X + \lambda I)^{-1/2}$. The computationally intensive part for this estimator is in evaluating the rank-r truncation $[\![B]\!]_r$. Its singular values and left singular vectors can be obtained by solving the symmetric eigenvalue problem $BB^* q_i = \sigma_i^2 q_i$. We rewrite $BB^*$

$$BB^* = P_Y \hat{\Phi}_{Y|X} \hat{\Phi}_X^* P_X (P_X \hat{\Phi}_X \hat{\Phi}_X^* P_X + n\lambda I)^{-1} P_X \hat{\Phi}_X \hat{\Phi}_{Y|X}^* P_Y$$

$$= P_Y \hat{\Phi}_{Y|X} \hat{\Phi}_X^* P_X \hat{\Phi}_X (\hat{\Phi}_X^* P_X \hat{\Phi}_X + n\lambda I)^{-1} \hat{\Phi}_{Y|X}^* P_Y$$

$$= P_Y \hat{\Phi}_{Y|X} K_{X,\tilde{X}} K_{\tilde{X},\tilde{X}}^\dagger K_{\tilde{X},X} (K_{X,\tilde{X}} K_{\tilde{X},\tilde{X}}^\dagger K_{\tilde{X},X} + n\lambda I)^{-1} \hat{\Phi}_{Y|X}^* P_Y$$

$$= P_Y \hat{\Phi}_{Y|X} K_{X,\tilde{X}} K_{\tilde{X},\tilde{X}}^\dagger (K_{\tilde{X},X} K_{X,\tilde{X}} K_{\tilde{X},\tilde{X}} + n\lambda I)^{-1} K_{\tilde{X},X} \hat{\Phi}_{Y|X}^* P_Y$$

$$= P_Y \hat{\Phi}_{Y|X} K_{X,\tilde{X}} (K_{\tilde{X},X} K_{X,\tilde{X}} + n\lambda K_{\tilde{X},\tilde{X}})^\dagger K_{\tilde{X},X} \hat{\Phi}_{Y|X}^* P_Y$$

where the second and fourth equalities are applications of the push-through identity, the third by definition of projections and kernel matrices, and the last by collecting $K_{\tilde{X},\tilde{X}}$. By construction, the non-trivial eigenfunctions of $BB^*$ are in the range of $P_Y \hat{\Phi}_{Y|X} K_{X,\tilde{X}}$, therefore we can set $q_i = P_Y \hat{\Phi}_{Y|X} K_{X,\tilde{X}} w_i$ for some $w_i \in \mathbb{R}^m$, and solve the following eigenvalue problem instead

$$P_Y \hat{\Phi}_{Y|X} K_{X,\tilde{X}} (K_{\tilde{X},X} K_{X,\tilde{X}} + n\lambda K_{\tilde{X},\tilde{X}})^\dagger K_{\tilde{X},X} \hat{\Phi}_{Y|X}^* P_Y \hat{\Phi}_{Y|X} K_{X,\tilde{X}} w_i = \sigma_i^2 P_Y \hat{\Phi}_{Y|X} K_{X,\tilde{X}} w_i$$

$$(K_{\tilde{X},X} K_{X,\tilde{X}} + n\lambda K_{\tilde{X},\tilde{X}})^\dagger K_{\tilde{X},X} K_{Y,\tilde{Y}} K_{\tilde{Y},\tilde{Y}}^\dagger K_{\tilde{Y},Y} K_{X,\tilde{X}} w_i = \sigma_i^2 w_i$$

where we have simplified the left term of both sides of the equation.

The eigenfunctions of $BB^*$ are therefore $q_i = P_Y \hat{\Phi}_{Y|X} K_{X,\tilde{X}} w_i$, which must be normalized as

$$\|q_i\|^2 = w_i^\top K_{\tilde{X},X} K_{Y,\tilde{Y}} K_{\tilde{Y},\tilde{Y}}^\dagger K_{\tilde{Y},Y} K_{X,\tilde{X}} w_i = 1.$$

Thanks to this normalization, the projector onto the $r$ leading left singular vectors of $B$ is $Q_r Q_r^*$, where $Q_r = [q_1, \dots, q_r]$. Then the NysRRR estimator can be written as

$$Q_r Q_r^* B (P_X \hat{C} P_X + \lambda I)^{-1/2}$$

where

$$B(P_X \hat{C} P_X + \lambda I)^{-1/2} = P_Y \hat{C}_{YX} P_X (P_X \hat{C} P_X + \lambda I)^{-1}$$

$$= P_Y \hat{\Phi}_{Y|X} K_{X,\tilde{X}} (K_{\tilde{X},X} K_{X,\tilde{X}} + n\lambda K_{\tilde{X},\tilde{X}})^{-1} \widetilde{\Phi}_X^*.$$

with the same techniques we used for rewriting $BB^*$. Finally, let $D_r$ and $E_r$ as in the statement. We can apply the projection to obtain

$$Q_r Q_r^* B (P_X \hat{\Phi}_X \hat{\Phi}_X^* P_X + n\lambda I)^{-1/2} = \widetilde{\Phi}_Y D_r E_r^* \widetilde{\Phi}_X^*.$$

$\square$

# D  Forecasting & Koopman Modes

The three estimators considered in Section C are all of the form
$$\hat{A}_\lambda = \widetilde{\Phi}_Y W \widetilde{\Phi}_X^*, \qquad W \in \mathbb{R}^{m \times m}.$$
We will use this generic form to provide expressions for the following operations:

1. producing forecasts of the dynamical system at a future time,
2. computing the approximate eigenvalues and eigenfunctions of the Koopman operator,
3. computing the Koopman modes.

## D.1  Forecasting

Given a new data-point $x \in \mathcal{X}$ and an observable function $g \in \mathcal{H}$ (note that this can simply be the identity function), we can approximate the one-step-ahead expectation $\mathbf{E}\big[g(X_{t+1})|X_t = x\big] =$

$(\mathcal{K}_\pi g)(\boldsymbol{x})$ by using the obtained estimators $\hat{A}^*$. Note that by the reproducing property $\widetilde{\Phi}_Y^* g = [g(\boldsymbol{y}_i), \ldots, g(\boldsymbol{y}_m)]^\top =: g_m$, then

$$(\hat{A}^* g)(x) = (\widetilde{\Phi}_X W^\top \widetilde{\Phi}_Y^* g)(\boldsymbol{x}) = (\widetilde{\Phi}_X W^\top g_m)(\boldsymbol{x}) = \sum_{i=1}^m (W^\top g_m)_i k(\tilde{\boldsymbol{x}}_i, \boldsymbol{x}).$$

### D.2 Eigenfunctions and eigenvalues

We wish to compute the eigenfunctions $\xi, \psi \in \mathcal{H}$, as well as the eigenvalues $\lambda_i$ of $\hat{A}$. The left eigenfunctions satisfy $\hat{A}^* \xi_i = \bar{\lambda}_i \xi_i$ and the right eigenfunctions satisfy $\hat{A} \psi_i = \lambda_i \psi_i$. In the following we will use Mollenhauer et al. [43, Proposition 3] to manipulate the eigendecomposition of operators in $\mathcal{H}$.

Consider the decomposition $W = U_r V_r^*$ with $U_r, V_r \in \mathbb{C}^{m \times r}$, which is available for all considered estimators with $r \leq m$. For example, in the Nyström RRR estimator of proposition C.4, we can simply take $U_r = D_r$ and $V_r = E_r$. For the Nyström KRR estimator instead, $r = m$ and we can take the whole of $W$ as our $U_r$ and $V_r = I$.

To compute the **right eigenfunctions** $\psi_i$, such that $(\widetilde{\Phi}_Y U_r V_r^* \widetilde{\Phi}_X^*)\psi_i = \lambda_i \psi_i$, consider the following equivalent eigendecomposition

$$V_r^* \widetilde{\Phi}_X^* \widetilde{\Phi}_Y U_r \tilde{g}_i = \lambda_i \tilde{g}_i, \quad \text{where } \psi_i = \widetilde{\Phi}_Y U_r \tilde{g}_i.$$

Note that $\widetilde{\Phi}_X^* \widetilde{\Phi}_Y = K_{\bar{X}, \tilde{Y}}$ is a finite-dimensional object which can easily be computed. The eigenfunctions $\psi_i$ must be normalized such that $\psi_i^* \psi_i = 1$ for every $i$, so we must have

$$\tilde{g}_i^* U_r^* \widetilde{\Phi}_Y^* \widetilde{\Phi}_Y U_r \tilde{g}_i = 1.$$

A very similar process can be followed to obtain the **left eigenfunctions** $\xi_i$, such that $\widetilde{\Phi}_X V_r U_r^* \widetilde{\Phi}_Y^* \xi_i = \bar{\lambda}_i \xi_i$. Here we consider instead

$$U_r^* \widetilde{\Phi}_Y^* \widetilde{\Phi}_X V_r \tilde{h}_i = \bar{\lambda}_i \tilde{h}_i, \quad \text{where } \xi_i = \widetilde{\Phi}_X V_r \tilde{h}_i.$$

where once again, $\widetilde{\Phi}_Y^* \widetilde{\Phi}_X = K_{\bar{X}, \tilde{Y}}^\top$ and the eigenfunctions must be normalized such that $\tilde{h}_i^* V_r^* \widetilde{\Phi}_X^* \widetilde{\Phi}_X V_r \tilde{h}_i = 1$ for every $i$. Finally, $\psi$ and $\xi$ must be orthogonal to each other: we must have for $i, j \in [r]$ that $\langle \psi_i, \bar{\xi}_j \rangle_\mathcal{H} = \delta_{ij}$ (where $\delta_{ij}$ is a Dirac delta equals to 1 when $i = j$ and 0 otherwise). We can compute

$$\langle \psi_i, \bar{\xi}_j \rangle_\mathcal{H} = \tilde{h}_i^* V_r^* K_{\bar{X}, \tilde{Y}} U_r \tilde{g}_i = \lambda_j \tilde{h}_i^* \tilde{g}_j,$$

and note that $\tilde{h}_i^* \tilde{g}_j = \delta_{ij}$, but we must normalize $\xi$ such that

$$\xi_i = \widetilde{\Phi}_X V_r \tilde{h}_i / \bar{\lambda}_i.$$

### D.3 Koopman modes

Given the eigendecomposition of any estimator $\hat{A}$ as $\hat{A}_r = \sum_{i=1}^r \lambda_i \psi_i \otimes \bar{\xi}_i$, for an observable $g$ we have the following

$$\hat{A}_r^* g = \sum_{i=1}^r \lambda_i \xi_i \langle g, \bar{\psi}_i \rangle_\mathcal{H}$$

where $\langle g, \bar{\psi}_i \rangle_\mathcal{H} = \gamma_i^g$ are the Koopman modes. Expanding the definition of $\psi_i$ we get

$$\gamma_i^g = \langle g, \bar{\psi}_i \rangle_\mathcal{H} = \tilde{g}_i^* U_r^* \widetilde{\Phi}_Y^* g = \tilde{g}_i^* U_r^* g_m \in \mathbb{C}^m$$

which we can efficiently compute.

## E Excess risk of the Nyström KRR estimator

### E.1 Almost-sure decomposition of the KRR excess risk

**Lemma E.1 (Excess risk decomposition in operator norm for KRR):** *Let Assumptions 4.1 to 4.3 and 4.5 hold. Then the Nyström KRR estimator* (8) *satisfies almost surely*

$$\mathcal{E}(\hat{A}_{m,\lambda}^{KRR})^{1/2} \leq a\lambda^{1/2} + a\theta_1^2\|(\hat{C}_\lambda - C_\lambda)C_\lambda^{-1/2}\|_{\mathcal{B}(\mathcal{H})} + \theta_1^2\|(C_{YX} - \hat{C}_{YX})C_\lambda^{-1/2}\|_{\mathcal{B}(\mathcal{H})}$$
$$+ a\theta_1\theta_2\theta_3\|P_X^\perp C_\lambda^{1/2}\|_{\mathcal{B}(\mathcal{H})} + \theta_1^2\|P_Y^\perp C_\lambda^{1/2}\|_{\mathcal{B}(\mathcal{H})}$$

*where* $\theta_1 := \|\hat{C}_\lambda^{-1/2}C_\lambda^{1/2}\|$, $\theta_2 := \|\hat{C}_\lambda^{1/2}C_\lambda^{-1/2}\|$, $\theta_3 := \|\hat{C}_\lambda^{-1}C_\lambda\|$, *and* $a$ *is the constant of Assumption 4.5.*

*Proof of Lemma E.1:*   Let $\theta_1 := \|\hat{C}_\lambda^{-1/2}C_\lambda^{1/2}\|$, $\theta_3 := \|\hat{C}_\lambda^{-1}C_\lambda\|$. As in Lemma C.1 define $g_{\mathrm{KRR}}(\hat{C}) := U(U^*\hat{C}U + \lambda I)^{-1}U^*$. We have

$$\begin{aligned}
\mathcal{E}(\hat{A}_{m,\lambda}^{\mathrm{KRR}})^{1/2} &= \|\Phi_{Y|X} - \hat{A}_{m,\lambda}^{\mathrm{KRR}}\Phi_X\|_{\mathcal{B}(L_\pi^2,\mathcal{H})} \\
&\leq \|\Phi_{Y|X} - A_\lambda\Phi_X\|_{\mathcal{B}(L_\pi^2,\mathcal{H})} + \|(A_\lambda - C_{YX}g_{\mathrm{KRR}}(\hat{C}))\Phi_X\|_{\mathcal{B}(L_\pi^2,\mathcal{H})} \\
&\quad + \|(C_{YX}g_{\mathrm{KRR}}(\hat{C}) - \hat{A}_{m,\lambda}^{\mathrm{KRR}})\Phi_X\|_{\mathcal{B}(L_\pi^2,\mathcal{H})} \\
&\leq \underbrace{\|\Phi_{Y|X} - A_\lambda\Phi_X\|_{\mathcal{B}(L_\pi^2,\mathcal{H})}}_{A} + \underbrace{\|(A_\lambda - C_{YX}g_{\mathrm{KRR}}(\hat{C}))C^{1/2}\|}_{B} \\
&\quad + \underbrace{\|(C_{YX}g_{\mathrm{KRR}}(\hat{C}) - \hat{A}_{m,\lambda}^{\mathrm{KRR}})C^{1/2}\|}_{C}
\end{aligned} \tag{32}$$

where we used the polar decomposition $\Phi_X^* = WC^{1/2}$ for some partial isometry $W : \mathcal{H} \to L_\pi^2$.
**The first term** is

$$\begin{aligned}
\|\Phi_{Y|X} - A_\lambda\Phi_X\| &= \|\Phi_X\mathcal{K}_\pi^* - C_{YX}C_\lambda^{-1}\Phi_X\| \\
&\leq a\lambda^{1/2} + \|(I - P_\mathcal{H})\Phi_{Y|X}\|
\end{aligned}$$

where we used the definition of $\Phi_{Y|X}$ and applied Lemma H.1.

**The second term** of our decomposition (32) can be bounded as follows:
It holds

$$\begin{aligned}
B &= \|C_{YX}(C_\lambda^{-1} - g_{\mathrm{KRR}}(\hat{C}))C^{1/2}\| \\
\text{(by Lemma H.2:)} \quad &\leq a\|C(C_\lambda^{-1} - g_{\mathrm{KRR}}(\hat{C}))C^{1/2}\| \\
&\leq a\left(\underbrace{\|C(C_\lambda^{-1} - \hat{C}_\lambda^{-1})C^{1/2}\|}_{B_1} + \underbrace{\|C(\hat{C}_\lambda^{-1} - g_{\mathrm{KRR}}(\hat{C}))C^{1/2}\|}_{B_2}\right)
\end{aligned}$$

We now bound the terms $B_1$ and $B_2$ separately.

$$\begin{aligned}
B_1 &= \|C(C_\lambda^{-1} - \hat{C}_\lambda^{-1})C^{1/2}\| \\
&= \|CC_\lambda^{-1}(\hat{C}_\lambda - C_\lambda)\hat{C}_\lambda^{-1}C^{1/2}\| \\
&\leq \|CC_\lambda^{-1}\|\|(\hat{C}_\lambda - C_\lambda)C_\lambda^{-1/2}\|\|C_\lambda^{1/2}\hat{C}_\lambda^{-1/2}\|\|\hat{C}_\lambda^{-1/2}C^{1/2}\| \\
&\leq \theta_1^2\|(\hat{C}_\lambda - C_\lambda)C_\lambda^{-1/2}\|
\end{aligned}$$

Let $\hat{P}_\lambda := \hat{C}_\lambda^{1/2}g_{\mathrm{KRR}}(\hat{C})\hat{C}_\lambda^{1/2}$. We recall that $g_{\mathrm{KRR}}(\hat{C}) = P_X g_{\mathrm{KRR}}(\hat{C})$, so that

$$\begin{aligned}
\hat{P}_\lambda^2 &= \hat{C}_\lambda^{1/2}(g_{\mathrm{KRR}}(\hat{C})\hat{C}_\lambda P_X)g_{\mathrm{KRR}}(\hat{C})\hat{C}_\lambda^{1/2} \\
&= \hat{C}_\lambda^{1/2}P_X g_{\mathrm{KRR}}(\hat{C})\hat{C}_\lambda^{1/2} \\
&= \hat{P}_\lambda.
\end{aligned}$$

This implies $\hat{P}_\lambda^2 = \hat{P}_\lambda = \hat{P}_\lambda^*$. Hence $\hat{P}_\lambda$ is an orthogonal projection, and defining $\hat{P}_\lambda^\perp = I - \hat{P}_\lambda$ it holds $\|\hat{P}_\lambda^\perp\|_{\mathcal{B}(\mathcal{H})} \leq 1$. We can thus bound $B_2$ as follows:

$$
\begin{aligned}
B_2 &= \|C(\hat{C}_\lambda^{-1} - g_{\mathrm{KRR}}(\hat{C}))C^{1/2}\| \\
&= \|C\hat{C}_\lambda^{-1/2}(I - \hat{P}_\lambda)\hat{C}_\lambda^{-1/2}C^{1/2}\| \\
\text{(by Lemma I.1)} \quad &= \|C\hat{C}_\lambda^{-1}P_X^\perp\hat{C}_\lambda^{1/2}(I - \hat{P}_\lambda)\hat{C}_\lambda^{-1/2}C^{1/2}\| \\
&= \|C\hat{C}_\lambda^{-1}\|_{\mathcal{B}(\mathcal{H})}\|P_X^\perp C_\lambda^{1/2}\|_{\mathcal{B}(\mathcal{H})}\|C_\lambda^{-1/2}\hat{C}_\lambda^{1/2}\|_{\mathcal{B}(\mathcal{H})}\|I - \hat{P}_\lambda\|\|\hat{C}_\lambda^{-1/2}C^{1/2}\|_{\mathcal{B}(\mathcal{H})} \\
&\leq \theta_1\theta_2\theta_3\|P_X^\perp C_\lambda^{1/2}\|
\end{aligned}
$$

**For the third term,** due to Lemma C.1:

$$
C = \|(C_{YX} - P_Y\hat{C}_{YX})g_{\mathrm{KRR}}(\hat{C})C^{1/2}\| \tag{33}
$$

$$
\leq \|(C_{YX} - P_Y\hat{C}_{YX})C_\lambda^{-1/2}\|\|C_\lambda^{1/2}\hat{C}_\lambda^{-1/2}\|_{\mathcal{B}(\mathcal{H})}\|\hat{P}_\lambda\|_{\mathcal{B}(\mathcal{H})}\|\hat{C}_\lambda^{-1/2}C^{1/2}\|_{\mathcal{B}(\mathcal{H})} \tag{34}
$$

$$
\leq \theta_1^2\|(C_{YX} - P_Y\hat{C}_{YX})C_\lambda^{-1/2}\|_{\mathcal{B}(\mathcal{H})} \tag{35}
$$

$$
\leq \theta_1^2\Big(\|P_Y^\perp C_\lambda^{1/2}\|_{\mathcal{B}(\mathcal{H})} + \|(C_{YX} - \hat{C}_{YX})C_\lambda^{-1/2}\|_{\mathcal{B}(\mathcal{H})}\Big) \tag{36}
$$

where we used Lemma J.7 for the last inequality.
Starting again from (32) and putting everything together, we get

$$
\begin{aligned}
\mathcal{E}(\hat{A}_{m,\lambda}^{\mathrm{KRR}})^{1/2} \leq \; & a\lambda^{1/2} \\
& + a\theta_1^2\|(\hat{C}_\lambda - C_\lambda)C_\lambda^{-1/2}\|_{\mathcal{B}(\mathcal{H})} \\
& + \theta_1^2\|(C_{YX} - \hat{C}_{YX})C_\lambda^{-1/2}\|_{\mathcal{B}(\mathcal{H})} \\
& + a\theta_1\theta_2\theta_3\|P_X^\perp C_\lambda^{1/2}\|_{\mathcal{B}(\mathcal{H})} \\
& + \theta_1^2\|P_Y^\perp C_\lambda^{1/2}\|_{\mathcal{B}(\mathcal{H})}.
\end{aligned}
$$

$\square$

## E.2 Excess risk rates for KRR

In order to control the terms appearing in our decomposition, we recall that Assumption 4.4 implies

$$
d_{\mathrm{eff}}(\lambda) \leq C_\beta\lambda^{-\beta} \quad \text{where} \quad C_\beta := \begin{cases} \dfrac{c}{1-\beta} & ,\beta < 1 \\ K^2 & ,\beta = 1 \end{cases}, \tag{37}
$$

where $c$ is the constant of Assumption 4.4, see [9, Proposition 3 with $b \to 1/\beta$ and $\beta \to c$] and [18, Lemma 11] which shows that the existence of a constant $C_\beta$ such that the first part of (37) holds implies in return $\lambda_i(C) \lesssim i^{-1/\beta}$.

*Proof of Theorem 4.6:* By Lemma E.1 taking $P_X = P_Y$, it holds almost surely

$$
\begin{aligned}
\mathcal{E}(\hat{A}_{m,\lambda}^{\mathrm{KRR}})^{1/2} \leq \; & a\lambda^{1/2} + a\theta_1^2\|(\hat{C}_\lambda - C_\lambda)C_\lambda^{-1/2}\|_{\mathcal{B}(\mathcal{H})} + \theta_1^2\|(C_{YX} - \hat{C}_{YX})C_\lambda^{-1/2}\|_{\mathcal{B}(\mathcal{H})} \\
& + a\theta_1\theta_2\theta_3\|P_X^\perp C_\lambda^{1/2}\|_{\mathcal{B}(\mathcal{H})} + \theta_1^2\|P_Y^\perp C_\lambda^{1/2}\|_{\mathcal{B}(\mathcal{H})}
\end{aligned}
$$

and we recall that $\theta_1 := \|\hat{C}_\lambda^{-1/2}C_\lambda^{1/2}\|$ and $\theta_3 := \|\hat{C}_\lambda^{-1}C_\lambda\|$. We bound separately the terms appearing in this expression.

**Bound of $\theta_1$ and $\theta_2$.** We control these term by bounding $\|C_\lambda^{-1/2}(\hat{C} - C)C_\lambda^{-1/2}\|$. By Lemma J.3 it holds for any $\delta' \in ]0, 1[$ and any $\lambda \in ]0, \|C\|_{\mathcal{B}(\mathcal{H})}]$ with probability $1 - \delta'$

$$
\left\|C_\lambda^{-1/2}(\hat{C} - C)C_\lambda^{-1/2}\right\| \leq \frac{4c_\tau\beta}{3n\lambda^\tau} + \sqrt{\frac{2c_\tau\beta}{n\lambda^\tau}} \quad \text{where} \quad \beta = \log\left(\frac{8K^2}{\delta'\lambda}\right) \tag{38}
$$

A sufficient condition to bound the right hand side of the previous expression by $1/4$ is to have $n\lambda^\tau > 32c_\tau\beta$ (in which cases both terms are bounded by $1/8$). Assuming this holds, $I - \|C_\lambda^{-1/2}(\hat{C}-$

$C)C_\lambda^{-1/2}\|$ is invertible and we also have

$$\theta_2^2 = \|\hat{C}_\lambda^{1/2}C_\lambda^{-1/2}\|^2 = \|C_\lambda^{-1/2}\hat{C}_\lambda C_\lambda^{-1/2}\| = \|I - C_\lambda^{-1/2}(C - \hat{C})C_\lambda^{-1/2}\|$$
$$\leq 1 + \|C_\lambda^{-1/2}(C - \hat{C})C_\lambda^{-1/2}\|$$
$$\leq 1.25$$
$$\text{and thus} \quad \theta_2 \leq 1.12$$
$$\text{while} \quad \theta_1^2 = \|\hat{C}_\lambda^{-1/2}C_\lambda^{1/2}\|^2 = \|(C_\lambda^{-1/2}\hat{C}_\lambda C_\lambda^{-1/2})^{-1}\|$$
$$\overset{(i)}{\leq} (1 - \|C_\lambda^{-1/2}(\hat{C} - C)C_\lambda^{-1/2}\|)^{-1}$$
$$\leq 1.34$$
$$\text{and thus} \quad \theta_1 \leq 1.16$$

where $(i)$ can be obtained by taking the Neumann expansion of $I - \|C_\lambda^{-1/2}(\hat{C} - C)C_\lambda^{-1/2}\|$.
**Bound for $\theta_3$.** By Lemma J.5 it holds with probability $1 - \delta'$

$$\|(C - \hat{C})C_\lambda^{-1}\|_{\text{op}} \leq \frac{2K\sqrt{c_\tau}\log(2/\delta')}{\lambda^{(\tau+1)/2}n} + \sqrt{\frac{2K^2\,\text{tr}(C_\lambda^{-2}C)\log(2/\delta')}{n}} \tag{39}$$

Both terms in the above rhs are bounded by $1/4$ provided

$$\lambda^{(\tau+1)/2}n \geq 8K\sqrt{c_\tau}\log(2/\delta')$$
$$n \geq 32K^2\lambda^{-(1+\beta)}\log(2/\delta')$$

where we used $\text{tr}(C_\lambda^{-2}C) = \sum\lambda_i(C)(\lambda_i(C) + \lambda)^{-2} \leq \lambda^{-1}\text{tr}(C_\lambda^{-1}C) \leq C_\beta\lambda^{-(1+\beta)}$. When this is the case, we have $\|(C - \hat{C})C_\lambda^{-1}\|_{\text{op}} \leq 1/2 < 1$ and the operator $I - (\hat{C}_\lambda - C_\lambda)C_\lambda^{-1}$ is invertible.

$$\theta_3 = \|(\hat{C}_\lambda C_\lambda^{-1})^{-1}\| = \|(I - (\hat{C}_\lambda - C_\lambda)C_\lambda^{-1})^{-1}\|$$
$$\overset{(i)}{\leq} (1 - \|(\hat{C}_\lambda - C_\lambda)C_\lambda^{-1}\|_{\mathcal{B}(\mathcal{H})})^{-1}$$
$$\leq 2.$$

where $(i)$ can be obtained by considering the Neumann expansion of $I - (\hat{C}_\lambda - C_\lambda)C_\lambda^{-1}$.
**Bound for $\|P_X^\perp C_\lambda^{1/2}\|$.** By Lemma J.6, provided $\lambda \in ]0, \|C\|_{\mathcal{B}(\mathcal{H})}]$ it holds with probability $1 - \delta'$

$$\|P_X^\perp C_\lambda^{1/2}\|_{\mathcal{B}(\mathcal{H})} \leq \sqrt{3\lambda}$$

provided $m \geq \max(67, 5\,\text{ess}\sup_{x\sim\pi}\|C_\lambda^{-1/2}\phi(x)\|^2)\log\frac{4K^2}{\lambda\delta'}$, which by Lemma H.3 is ensured if $m \geq \max(67, 5\frac{c_\tau}{\lambda^\tau})\log\frac{4K^2}{\lambda\delta'}$.
**Bound for $\|(C - \hat{C})C_\lambda^{-1/2}\|$ and $\|(C_{YX} - \hat{C}_{YX})C_\lambda^{-1/2}\|$.** By Lemma J.4, for any $\delta' \in ]0, 1[$, each of the following events holds with probability $1 - 2\delta'$:

$$\max(\|(C - \hat{C})C_\lambda^{-1/2}\|, \|(C_{YX} - \hat{C}_{YX})C_\lambda^{-1/2}\|) \leq \frac{2K\sqrt{c_\tau}\log(2/\delta')}{\lambda^{\tau/2}n} + \sqrt{\frac{2K^2 d_{\text{eff}}(\lambda)\log(2/\delta')}{n}} \tag{40}$$

By Eq. (37) we have $d_{\text{eff}}(\lambda) \leq C_\beta\lambda^{-\beta}$.
Choosing $\delta' = \delta/5$, we get via a union bound with probability $1 - \delta$ that $\theta_1\theta_2\theta_3 \leq 2.6$, $\theta_1^2 \leq 1.34$ and

$$\mathcal{E}(\hat{A}_{m,\lambda}^{\text{KRR}})^{1/2} \leq a\lambda^{1/2} + 1.34(a + 1)\left(\frac{2K\sqrt{c_\tau}\log(2/\delta')}{\lambda^{\tau/2}n} + \sqrt{\frac{2K^2 C_\beta\log(2/\delta')}{n\lambda^\beta}}\right) + (2.6a + 1.34)\sqrt{3}\lambda^{1/2}$$

$$\leq c_1\lambda^{1/2} + c_2\lambda^{-\tau/2}n^{-1} + c_3\lambda^{-\beta/2}n^{-1/2}$$
$$\text{where:} \quad c_1 := (5.5a + 2.33)$$
$$c_2 := 1.34(a + 1)2K\sqrt{c_\tau}\log(2/\delta')$$
$$c_3 := 1.34(a + 1)\sqrt{2K^2 C_\beta\log(2/\delta')}$$

for any $\lambda$ and $m$ satisfying the constraints

$$
\begin{cases}
\lambda > n^{-1/\tau} (32 c_\tau)^{1/\tau} \log\left(\frac{8K^2}{\delta'\lambda}\right)^{1/\tau} \\
\lambda \geq n^{-2/(\tau+1)} (8K\sqrt{c_\tau} \log(2/\delta'))^{2/(\tau+1)} \\
\lambda \geq n^{-1/(1+\beta)} (32K^2 \log(2/\delta'))^{1/(1+\beta)} \\
\lambda \in ]0, K^2]. \\
m \geq \max(67, 5\frac{c_\tau}{\lambda^\tau}) \log \frac{4K^2}{\lambda\delta'} \quad \text{(uniform sampling)}
\end{cases}
\tag{41}
$$

We pick $\boxed{\lambda := c_\lambda n^{-1/(1+\beta)}}$ which is asymptotically the saturating constraint (given that $1/(1+\beta) < 1 < 2/(\tau+1) \leq 1/\tau$), where $c_\lambda$ is a constant choosen to enforce the following equations (which are sufficient conditions for eq. (41) to hold):

$$
\begin{cases}
c_\lambda^\tau n^{1-\tau/(1+\beta)} > (32 c_\tau) \log\left(\frac{8K^2 n^{1/(1+\beta)}}{\delta' c_\lambda}\right) \\
c_\lambda^{(\tau+1)/2} n^{1-(\tau+1)/(2(1+\beta))} \geq 8K\sqrt{c_\tau} \log(2/\delta') \\
\qquad\qquad\quad c_\lambda \geq (32K^2 \log(2/\delta'))^{1/(1+\beta)} \\
c_\lambda n^{-1/(1+\beta)} \leq K^2
\end{cases}
\tag{42}
$$

As $1 - (\tau+1)/(2(1+\beta)) > 0$, a sufficient condition for the second equation is

$$
c_\lambda \geq (8K\sqrt{c_\tau} \log(2/\delta'))^{2/(\tau+1)}.
$$

Assuming $c_\tau \geq 8K^2$, a sufficient condition to satisfy the first constraint is

$$
c_\lambda^\tau n^{1-\tau/(1+\beta)} > (32 c_\tau) 2 \max(\log\left(n^{1/(1+\beta)}\right), \log\left((\delta')^{-1}\right))
$$

which is in particular ensured (noting that $\log(n)/n^\nu \leq 1/(\nu e)$ for any $n, \nu > 0$) whenever

$$
c_\lambda > (64 c_\tau \max((e(1+\beta-\tau))^{-1}, \log(1/\delta')))^{1/\tau}
$$

Noting that $1 + \beta - \tau \leq 1$, we get that

$$
c_\lambda > (64 c_\tau (e(1+\beta-\tau))^{-1} \log(1/\delta'))^{1/\tau}
$$

is also sufficient. We recall that $1/(1+\beta) < 1 < 2/(\tau+1) \leq 1/\tau$, so that we can choose

$$
\boxed{c_\lambda := \log(2/\delta')^{1/\tau} \max((32K^2)^{1/(1+\beta)}, (8K\sqrt{c_\tau})^{2/(\tau+1)}, (64 c_\tau (e(1+\beta-\tau))^{-1})^{1/\tau}, 8K^2)}
$$

while the last constraint $n \geq (c_\lambda/K^2)^{1+\beta}$ is satisfied by assumption.

$$
\begin{aligned}
\mathcal{E}(\hat{A}_{m,\lambda}^{\text{KRR}})^{1/2} &\leq c_1 \lambda^{1/2} + c_2 \lambda^{-\tau/2} n^{-1} + c_3 \lambda^{-\beta/2} n^{-1/2} \\
&\leq c_1 c_\lambda^{1/2} n^{-1/(2(1+\beta))} + c_2 c_\lambda^{-\tau/2} n^{\tau/(2(1+\beta))-1} + c_3 c_\lambda^{-\beta/2} n^{\beta/(2(1+\beta))-1/2} \\
&\leq c_1 c_\lambda^{1/2} n^{-1/(2(1+\beta))} + c_2 c_\lambda^{-\tau/2} n^{-(1+2\beta+(1-\tau))/(2(1+\beta))} + c_3 c_\lambda^{-\beta/2} n^{-1/(2(1+\beta))} \\
&\leq (c_1 c_\lambda^{1/2} + c_2 c_\lambda^{-\tau/2} + c_3 c_\lambda^{-\beta/2}) n^{-1/(2(1+\beta))}.
\end{aligned}
$$

which gives the claimed result. The last constraint (on $m$) is satisfied by the assumptions of the lemma. $\qquad\square$

# F  Excess risk of the Nyström RRR estimator

Recalling (30), NyströmRRR estimator is of the form $\hat{A}_{m,\lambda}^{\text{RRR}} = [\![\tilde{B}]\!]_r (\tilde{C}_\lambda)^{-1/2}$, where $\tilde{B} := \tilde{C}_{YX}(\tilde{C}_\lambda)^{-1/2}$ for $\tilde{C}_{YX} := P_Y \hat{C}_{YX} P_X$ and $\tilde{C}_\lambda := P_X \hat{C} P_X + \lambda I$. While the population version is $A_\lambda^{\text{RRR}} := [\![B]\!]_r C_\lambda^{-1/2}$ where $B := C_{YX}(C_\lambda)^{-1/2}$.

In this section we follow the approach in [30] and decompose the operator norm excess risk in the following way:

$$\mathcal{E}(\hat{A}_{m,\lambda}^{\text{RRR}})^{1/2} = \|\Phi_{Y|X} - A_\lambda \Phi_X\|_{\mathcal{B}(L_\pi^2,\mathcal{H})} + \|(A_\lambda - A_\lambda^{\text{RRR}})\Phi_X\|_{\mathcal{B}(L_\pi^2,\mathcal{H})} + \|(A_\lambda^{\text{RRR}} - \hat{A}_{m,\lambda}^{\text{RRR}})\Phi_X\|_{\mathcal{B}(L_\pi^2,\mathcal{H})}$$

Then, recalling that $A_\lambda = C_{YX} C_\lambda^{-1}$ and $\hat{A}_{m,\lambda}^{\text{KRR}} = \tilde{B} \tilde{C}_\lambda^{-1/2}$, we also have $A_\lambda^{\text{RRR}} = P_B A_\lambda$ and $\hat{A}_{m,\lambda}^{\text{RRR}} = P_{\tilde{B}} \hat{A}_{m,\lambda}^{\text{KRR}}$, where $P_B$ and $P_{\tilde{B}}$ are orthogonal projectors onto leading $r$ left singular vectors of $B$ and $\tilde{B}$, respectively.

Thus,

$$\begin{aligned}
\mathcal{E}(\hat{A}_{m,\lambda}^{\text{RRR}})^{1/2} &\le a\,\lambda^{1/2} + \sigma_{r+1}(\Phi_{Y|X}) + \|(A_\lambda^{\text{RRR}} - \hat{A}_{m,\lambda}^{\text{RRR}})\Phi_X\|_{\mathcal{B}(\mathcal{H})} \\
&= a\,\lambda^{1/2} + \sigma_{r+1}(\Phi_{Y|X}) + \|(P_B A_\lambda - P_{\tilde{B}} \hat{A}_{m,\lambda}^{\text{KRR}})\Phi_X\|_{\mathcal{B}(\mathcal{H})} \\
&\le a\,\lambda^{1/2} + \sigma_{r+1}(\Phi_{Y|X}) + \|((P_B - P_{\tilde{B}})A_\lambda \Phi_X\|_{\mathcal{B}(\mathcal{H})} + \|P_{\tilde{B}}(A_\lambda - \hat{A}_{m,\lambda}^{\text{KRR}})\Phi_X\|_{\mathcal{B}(\mathcal{H})} \\
&\le a\,\lambda^{1/2} + \sigma_{r+1}(\Phi_{Y|X}) + K\frac{\|\tilde{B}\tilde{B}^* - BB^*\|_{\mathcal{B}(\mathcal{H})}}{\sigma_r^2(B) - \sigma_{r+1}^2(B)} + \|(A_\lambda - \hat{A}_{m,\lambda}^{\text{KRR}})\Phi_X\|_{\mathcal{B}(\mathcal{H})}
\end{aligned}$$

where the last inequality is due to $\|A_\lambda\| \le a$ and [30, Proposition 4].

Recalling Lemma E.1, we observe that

$$\mathcal{E}(\hat{A}_{m,\lambda}^{\text{RRR}})^{1/2} \le \sigma_{r+1}(\Phi_{Y|X}) + K\frac{\|\tilde{B}\tilde{B}^* - BB^*\|}{\sigma_r^2(B) - \sigma_{r+1}^2(B)} + \underbrace{a\,\lambda^{1/2} + \|(A_\lambda - \hat{A}_{m,\lambda}^{\text{KRR}})\Phi_X\|}_{\le \text{r.h.s. of the bound in Lemma E.1}}$$

Therefore, to prove Lemma 4.7 for the RRR estimator we just need to bound $\|\tilde{B}\tilde{B}^* - BB^*\|_{\mathcal{B}(\mathcal{H})}$. To that end, observe that, after some algebra, one obtains

$$\tilde{B}\tilde{B}^* - BB^* = A_\lambda(\tilde{C}_{YX} - C_{YX})^* + (\tilde{C}_{YX} - C_{YX})A_\lambda^* - A_\lambda(\tilde{C}_\lambda - C_\lambda)A_\lambda^* + (\tilde{A}_\lambda - A_\lambda)\tilde{C}_\lambda(\tilde{A}_\lambda - A_\lambda)^*,$$

and, consequently,

$$\begin{aligned}
\|\tilde{B}\tilde{B}^* - BB^*\|_{\mathcal{B}(\mathcal{H})} \le\; & 2a\|\tilde{C}_{YX} - C_{YX}\|_{\mathcal{B}(\mathcal{H})} + a^2\|P_X \hat{C} P_X - C\|_{\mathcal{B}(\mathcal{H})} \\
& + \|C_\lambda^{-1/2}\tilde{C}_\lambda C_\lambda^{-1/2}\|_{\mathcal{B}(\mathcal{H})}\|(\tilde{A}_\lambda - A_\lambda)C_\lambda^{1/2}\|_{\mathcal{B}(\mathcal{H})}^2,
\end{aligned}$$

follows using that $\|A_\lambda\|_{\mathcal{B}(\mathcal{H})} \le a$.

On the other hand,

$$\|\tilde{C}_{YX} - C_{YX}\|_{\mathcal{B}(\mathcal{H})} \le \|P_Y(\hat{C}_{YX} - C_{YX})P_X\|_{\mathcal{B}(\mathcal{H})} + \|P_Y^\perp C_{YX} P_X\|_{\mathcal{B}(\mathcal{H})} + \|C_{YX} P_X^\perp\|_{\mathcal{B}(\mathcal{H})},$$

which implies that

$$\|\tilde{C}_{YX} - C_{YX}\|_{\mathcal{B}(\mathcal{H})} \le \|\hat{C}_{YX} - C_{YX}\|_{\mathcal{B}(\mathcal{H})} + 2\,a\,K\,\varepsilon_1,$$

where $\varepsilon_1 := \max\{\|P_X^\perp C^{1/2}\|_{\mathcal{B}(\mathcal{H})}, \|P_Y^\perp C^{1/2}\|_{\mathcal{B}(\mathcal{H})}\}$. Similarly, we obtain

$$\|\tilde{C}_\lambda - C_\lambda\|_{\mathcal{B}(\mathcal{H})} \le \|\hat{C} - C\|_{\mathcal{B}(\mathcal{H})} + 2\,K\,\varepsilon_1. \tag{43}$$

But, $\varepsilon_1$ can be bounded by Lemma J.6. Indeed, provided $\lambda \in\,]0, \|C\|_{\mathcal{B}(\mathcal{H})}]$, it holds with probability $1 - \delta'$

$$\varepsilon_1 \le \sqrt{3\lambda}$$

provided $m \ge \max(67, 5\operatorname{ess\,sup}_{x\sim\pi}\|C_\lambda^{-1/2}\phi(x)\|^2)\log\frac{4K^2}{\lambda\delta'}$, which by Lemma H.3 is ensured if $m \ge \max(67, 5\frac{c_\tau}{\lambda^\tau})\log\frac{4K^2}{\lambda\delta'}$.

Additionally,

$$\|C_\lambda^{-1/2}\tilde{C}_\lambda C_\lambda^{-1/2}\|_{\mathcal{B}(\mathcal{H})} \le \|C_\lambda^{-1/2}P_X\hat{C}_\lambda P_X C_\lambda^{-1/2}\|_{\mathcal{B}(\mathcal{H})} + \lambda\|C_\lambda^{-1/2}P_X^\perp C_\lambda^{-1/2}\|_{\mathcal{B}(\mathcal{H})}$$
$$\le \theta_2^2\|\hat{C}_\lambda^{-1/2}P_X\hat{C}_\lambda P_X C_\lambda^{-1/2}\|_{\mathcal{B}(\mathcal{H})} + 1$$
$$\le \theta_2^2\|\hat{C}_\lambda^{1/2}P_X\hat{C}_\lambda^{-1}P_X C_\lambda^{1/2}\|_{\mathcal{B}(\mathcal{H})} + 1$$
$$\le \theta_2^2\|\hat{C}_\lambda^{1/2}P_X(P_X\hat{C}_\lambda P_X)^\dagger P_X C_\lambda^{1/2}\|_{\mathcal{B}(\mathcal{H})} + 1,$$

implies that

$$\|C_\lambda^{-1/2}\tilde{C}_\lambda C_\lambda^{-1/2}\|_{\mathcal{B}(\mathcal{H})} \le \theta_2^2 + 1 \le 2.25, \tag{44}$$

provided, as above, that $n\lambda^\tau > 32c_\tau\beta$.

Therefore, setting $\varepsilon_0 := \max\{a\|\hat{C}_{YX} - C_{YX}\|_{\mathcal{B}(\mathcal{H})}, a^2\|\hat{C} - C\|_{\mathcal{B}(\mathcal{H})}\}$, for all $i \in [m]$ we have

$$\left|\sigma_i^2(\tilde{B}) - \sigma_i^2(B)\right| \le \|\tilde{B}\tilde{B}^* - BB^*\| \le 3\varepsilon_0 + 6.93\,K\,a^2\lambda^{1/2} + 2.25\,\varepsilon_2^2, \tag{45}$$

where $\varepsilon_2 := \|(A_\lambda - \hat{A}_{m,\lambda}^{\mathrm{KRR}})C_\lambda^{1/2}\|$ is the variance of Nyström KRR estimator, and conclude that

$$\mathcal{E}(\hat{A}_{m,\lambda}^{\mathrm{RRR}})^{1/2} \le \sigma_{r+1}(\Phi_{Y|X}) + K\,\frac{3\varepsilon_0 + 6.93\,K\,a^2\,\lambda^{1/2} + 2.25\,\varepsilon_2^2}{\sigma_r^2(B) - \sigma_{r+1}^2(B)} + a\,\lambda^{1/2} + \varepsilon_2.$$

Therefore, the proof of Lemma 4.7 for RRR estimator directly follows from the bound on $a\,\lambda^{1/2} + \varepsilon_2$ given in the proof of Theorem 4.6, and the fact that, see e.g. [29], $\varepsilon_0 \lesssim n^{-1/2} \lesssim \lambda^{1/2}$.

## G  Excess risk of the Nyström PCR estimator

Recalling Eq. (27), NyströmPCR estimator is of the form

$$\hat{A}_m^{\mathrm{PCR}} = P_Y\hat{C}_{YX}[\![P_X\hat{C}P_X]\!]_r^\dagger = \tilde{C}_{YX}[\![\tilde{C}_\lambda]\!]_r = \hat{A}_{m,\lambda}^{\mathrm{KRR}}P_{\tilde{C}_\lambda},$$

for $\lambda = 0$ and with $P_{\tilde{C}_\lambda}$ being the orthogonal projector onto leading $r$ eigenspace of $\tilde{C}_\lambda$. So, to prove Lemma 4.7 for PCR estimator, denote $\hat{A}_{m,\lambda}^{\mathrm{PCR}} := \hat{A}_{m,\lambda}^{\mathrm{KRR}}P_{\tilde{C}_\lambda}$ for $\lambda \ge 0$, and let us define the population version $A_\lambda^{\mathrm{PCR}} = A_\lambda P_{C_\lambda}$, where $P_{C_\lambda}$ being the orthogonal projector onto leading $r$ eigenspace of $C_\lambda$.

As in the previous section we start with decomposition

$$\mathcal{E}(\hat{A}_m^{\mathrm{PCR}})^{1/2} = \|\Phi_{Y|X} - A_\lambda\Phi_X\|_{\mathcal{B}(L_\pi^2,\mathcal{H})} + \|(A_\lambda - A_\lambda^{\mathrm{PCR}})\Phi_X\|_{\mathcal{B}(L_\pi^2,\mathcal{H})} +$$
$$\|(A_\lambda^{\mathrm{PCR}} - \hat{A}_{m,\lambda}^{\mathrm{PCR}})\Phi_X\|_{\mathcal{B}(L_\pi^2,\mathcal{H})} + \|(\hat{A}_{m,\lambda}^{\mathrm{PCR}} - \hat{A}_m^{\mathrm{PCR}})\Phi_X\|_{\mathcal{B}(L_\pi^2,\mathcal{H})}.$$

The first and the second term are easily bounded by $\|\Phi_{Y|X} - A_\lambda\Phi_X\|_{\mathcal{B}(L_\pi^2,\mathcal{H})} \le a\,\lambda^{1/2}$, and

$$\|(A_\lambda - A_\lambda^{\mathrm{PCR}})\Phi_X\|_{\mathcal{B}(L_\pi^2,\mathcal{H})} = \|A_\lambda(I - P_{C_\lambda})\Phi_X\|_{\mathcal{B}(L_\pi^2,\mathcal{H})}$$
$$\le a\,\|(I - P_{C_\lambda})C^{1/2}\|_{\mathcal{B}(\mathcal{H})} \le a\,\sigma_{r+1}(\Phi_X).$$

For the third term, start by observing that

$$\|(A_\lambda^{\mathrm{PCR}} - \hat{A}_{m,\lambda}^{\mathrm{PCR}})\Phi_X\|_{\mathcal{B}(L_\pi^2,\mathcal{H})} = \|(A_\lambda P_{C_\lambda} - \hat{A}_{m,\lambda}^{\mathrm{KRR}}P_{\tilde{C}_\lambda})C^{1/2}\|_{\mathcal{B}(\mathcal{H})}$$
$$\le \|A_\lambda(P_{C_\lambda} - P_{\tilde{C}_\lambda})C^{1/2}\|_{\mathcal{B}(\mathcal{H})} + \|(A_\lambda - \hat{A}_{m,\lambda}^{\mathrm{KRR}})P_{\tilde{C}_\lambda}C^{1/2}\|_{\mathcal{B}(\mathcal{H})}$$
$$\le a\,K\,\|P_{C_\lambda} - P_{\tilde{C}_\lambda}\|_{\mathcal{B}(\mathcal{H})} + \|(A_\lambda - \hat{A}_{m,\lambda}^{\mathrm{KRR}})P_{\tilde{C}_\lambda}C^{1/2}\|_{\mathcal{B}(\mathcal{H})}$$
$$\le a\,K\,\frac{\|\tilde{C}_\lambda - C_\lambda\|_{\mathcal{B}(\mathcal{H})}}{\sigma_r^2(\Phi_X) - \sigma_{r+1}^2(\Phi_X)} + \|(A_\lambda - \hat{A}_{m,\lambda}^{\mathrm{KRR}})P_{\tilde{C}_\lambda}C^{1/2}\|_{\mathcal{B}(\mathcal{H})}$$
$$\le K\,\frac{\varepsilon_0 + 2\,a\,K\,\varepsilon_1}{\sigma_r^2(\Phi_X) - \sigma_{r+1}^2(\Phi_X)} + \|(A_\lambda - \hat{A}_{m,\lambda}^{\mathrm{KRR}})P_{\tilde{C}_\lambda}C^{1/2}\|_{\mathcal{B}(\mathcal{H})}$$

where the second last inequality is due to [30, Proposition 4] and the last one uses Eq. (43). Moreover, we have that

$$\|(A_\lambda - \hat{A}^{\text{KRR}}_{m,\lambda})P_{\tilde{C}_\lambda}C^{1/2}\|_{\mathcal{B}(\mathcal{H})} \le \|(A_\lambda - \hat{A}^{\text{KRR}}_{m,\lambda})C^{1/2}_\lambda\|_{\mathcal{B}(\mathcal{H})} \|C^{-1/2}_\lambda \tilde{C}^{1/2}_\lambda\|_{\mathcal{B}(\mathcal{H})} \|\tilde{C}^{-1/2}_\lambda P_{\tilde{C}_\lambda}C^{1/2}\|_{\mathcal{B}(\mathcal{H})}$$

$$\le \varepsilon_2 \sqrt{1+\theta_2^2} \,\|P_{\tilde{C}_\lambda}\tilde{C}^{-1/2}_\lambda C^{1/2}\|_{\mathcal{B}(\mathcal{H})}$$

$$\le \varepsilon_2 \sqrt{1+\theta_2^2} \,\|C^{1/2}_\lambda \tilde{C}^{-1/2}_\lambda\|_{\mathcal{B}(\mathcal{H})},$$

where we have used Eq. (44) and the fact that $P_{\tilde{C}_\lambda}$ is the spectral projector of $\tilde{C}^{-1/2}_\lambda$. Therefore, due to

$$\|C^{1/2}_\lambda \tilde{C}^{-1/2}_\lambda\|_{\mathcal{B}(\mathcal{H})} = \|C^{1/2}_\lambda[P^\perp_X + P_X]\tilde{C}^{-1/2}_\lambda\|_{\mathcal{B}(\mathcal{H})} \le \|C^{1/2}_\lambda P_X \tilde{C}^{-1/2}_\lambda\|_{\mathcal{B}(\mathcal{H})} + \|C^{1/2}_\lambda P^\perp_X \tilde{C}^{-1/2}_\lambda\|_{\mathcal{B}(\mathcal{H})}$$

$$\le \theta_2 \|\hat{C}^{1/2}_\lambda P_X \tilde{C}^{-1/2}_\lambda\|_{\mathcal{B}(\mathcal{H})} + \|C^{1/2}_\lambda P^\perp_X\|_{\mathcal{B}(\mathcal{H})}\|\tilde{C}^{-1/2}_\lambda\|_{\mathcal{B}(\mathcal{H})}$$

$$\le \theta_2 \|\hat{C}^{1/2}_\lambda P_X (P_X \hat{C}_\lambda P_X)^\dagger P_X \hat{C}^{1/2}_\lambda\|^{1/2}_{\mathcal{B}(\mathcal{H})} + \|C^{1/2}_\lambda P^\perp_X\|_{\mathcal{B}(\mathcal{H})}\lambda^{-1/2}$$

$$\le \theta_2 + \varepsilon_1 \lambda^{-1/2}$$

we obtain

$$\|(A_\lambda - \hat{A}^{\text{KRR}}_{m,\lambda})P_{\tilde{C}_\lambda}C^{1/2}\|_{\mathcal{B}(\mathcal{H})} \le \varepsilon_2 \left(1.68 + 1.5\lambda^{-1/2}\varepsilon_1\right),$$

provided that $n\lambda^\tau > 32c_\tau\beta$.

Finally for the last term, observe that $[\![\tilde{C}_\lambda]\!]^\dagger_r$ and $[\![\tilde{C}_0]\!]^\dagger_r$ share the same eigenvectors, and hence $[\![\tilde{C}_0]\!]^\dagger_r - [\![\tilde{C}_\lambda]\!]^\dagger_r = \lambda[\![\tilde{C}_\lambda \tilde{C}_0]\!]^\dagger_r$. Hence, it holds that

$$\|(\hat{A}^{\text{PCR}}_{m,\lambda} - \hat{A}^{\text{PCR}}_m)\Phi_X\|_{\mathcal{B}(L^2_\pi, \mathcal{H})} = \|\tilde{C}_{YX}([\![\tilde{C}_\lambda]\!]^\dagger_r - [\![\tilde{C}_0]\!]^\dagger_r)C^{1/2}\|_{\mathcal{B}(\mathcal{H})} = \lambda\|\tilde{C}_{YX}\tilde{C}^{-1}_\lambda[\![\tilde{C}_0]\!]^\dagger_r C^{1/2}\|_{\mathcal{B}(\mathcal{H})}$$

$$\le \lambda\|\tilde{C}_{YX}\tilde{C}^{-1}_\lambda[\![\tilde{C}_0]\!]^\dagger_r\tilde{C}^{1/2}_\lambda\|_{\mathcal{B}(\mathcal{H})}\|\tilde{C}^{-1/2}_\lambda C^{1/2}_\lambda\|_{\mathcal{B}(\mathcal{H})}$$

$$= \lambda\|\tilde{C}_{YX}\tilde{C}^{-1/2}_\lambda[\![\tilde{C}_0]\!]^\dagger_r\|_{\mathcal{B}(\mathcal{H})}\|\tilde{C}^{-1/2}_\lambda C^{1/2}_\lambda\|_{\mathcal{B}(\mathcal{H})}$$

$$= \lambda \,\|[\![\tilde{C}_0]\!]^\dagger_r\|_{\mathcal{B}(\mathcal{H})} \,\|\tilde{B}\|_{\mathcal{B}(\mathcal{H})} \,\|\tilde{C}^{-1/2}_\lambda C^{1/2}_\lambda\|_{\mathcal{B}(\mathcal{H})}.$$

Now, recalling Eq. (45), we can bound

$$\|\tilde{B}\|^2_{\mathcal{B}(\mathcal{H})} \le \|C^{-1/2}_\lambda C^*_{YX}C_{YX}C^{-1/2}_\lambda\|_{\mathcal{B}(\mathcal{H})} + \|BB^* - \tilde{B}\tilde{B}^*\|_{\mathcal{B}(\mathcal{H})}$$

$$\le a^2 K^2 + 3\varepsilon_0 + 6.93\, K\, a^2 \lambda^{1/2} + 2.25\,\varepsilon_2^2,$$

and,

$$\lambda^{1/2}\|[\![\tilde{C}_0]\!]^\dagger_r\|_{\mathcal{B}(\mathcal{H})} = \frac{\lambda^{1/2}}{\lambda_r(P_X\hat{C}P_X)} \le \frac{\lambda^{1/2}}{\lambda_r(C) - \|C_\lambda - \tilde{C}_\lambda\|_{\mathcal{B}(\mathcal{H})}} \le \frac{\lambda^{1/2}}{\sigma^2_r(\Phi_X) - \|\hat{C} - C\|_{\mathcal{B}(\mathcal{H})} - 2\,K\,\varepsilon_1}.$$

Thus, consequently, we obtain

$$\|(\hat{A}^{\text{PCR}}_{m,\lambda} - \hat{A}^{\text{PCR}}_m)\Phi_X\|_{\mathcal{B}(L^2_\pi, \mathcal{H})} \le \lambda^{1/2}\left(\theta_2 + \varepsilon_1\lambda^{-1/2}\right)\left(a^2 K^2 + 3\varepsilon_0 + 6.93\, K\, a^2\lambda^{1/2} + 2.25\,\varepsilon_2^2\right) \cdot$$

$$\frac{\lambda^{1/2}}{\sigma^2_r(\Phi_X) - \|\hat{C} - C\|_{\mathcal{B}(\mathcal{H})} - 2\,K\,\varepsilon_1}.$$

To conclude, observe that $r > n^{\frac{1}{\beta(1+\beta)}}$ due to Assumption 4.4 implies that $\sigma_{r+1}(\Phi_X) \lesssim n^{-\frac{1}{2(1+\beta)}}$. Therefore, collecting all the terms, under the assumptions of Lemma 4.7 we obtain

$$\mathcal{E}(\hat{A}^{\text{PCR}}_m)^{1/2} \lesssim c_{\text{PCR}}\, n^{-\frac{1}{2(1+\beta)}},$$

where $c_{\text{PCR}} = (\sigma^2_r(\Phi_X) - \sigma^2_{r+1}(\Phi_X))^{-1}$ is the problem dependant constant.

## H   Auxiliary results

**Lemma H.1 ([30, Proposition 2] with $\alpha = 1$):** *Under Assumption 4.2 it holds*

$$\|\Phi_X \mathcal{K}_\pi^* - C_{YX} C_\lambda^{-1} \Phi_X\| \leq a\lambda^{1/2}.$$

**Lemma H.2:** *Let $A$ be a bounded operator. Under Assumptions 4.1 and 4.5, it holds*

$$\|C_{YX} A\| \leq a\|CA\| \tag{46}$$

*Proof of Lemma H.2:* Note that under Assumption 4.5, as $C_{XY} C_{YX} \preccurlyeq a^2 C^2$ it also holds $A^* C_{XY} C_{YX} A \preccurlyeq a^2 A C^2 A$ and thus:

$$\begin{aligned}
\|C_{YX} A\| &= \|A^* C_{YX} C_{YX} A\|^{1/2} \\
&\leq a\|A^* C^2 A\|^{1/2} \\
&= a\|CA\|.
\end{aligned}$$

$\square$

The next lemma is a consequence of Assumption 4.3 and will be used in our concentration inequalities.

**Lemma H.3:** *Under Assumption 4.3, it holds $\pi$-almost surely for any $\nu$:*

$$\left\| C_\lambda^{-(1-\nu)/2} \phi(x) \right\|^2 \leq c_\tau \lambda^{-[\tau-\nu]_+} K^{2[\nu-\tau]_+}.$$

*The two following corollaries can be obtained picking $\nu = 0$ and $\nu = -1$:*

$$\left\| C_\lambda^{-1/2} \phi(x) \right\|^2 \leq \frac{c_\tau}{\lambda^\tau} \quad and \quad \left\| C_\lambda^{-1} \phi(x) \right\|^2 \leq \frac{c_\tau}{\lambda^{\tau+1}}.$$

*Proof of Lemma H.3:* By [18, Theorem 9], it holds $c_\tau := \|k_\pi^\tau\|_\infty^2 = \operatorname{ess\,sup}_{x \sim \pi} \sum_{i \in I} \mu_i^\tau |e_i(x)|^2$ (where $(e_i)$ is defined in Section A.3, and we recall that $(\sqrt{\mu_i} e_i)_{i \in \mathbb{N}}$ is an orthonormal basis of $\mathcal{H}$. Denoting $\mu_i := \lambda_i(C)$, it holds

$$\begin{aligned}
\left\| C_\lambda^{-(1-\nu)/2} \phi(x) \right\|^2 &= \left\| \left( \sum_{i \in I} (\mu_i + \lambda)^{-(1-\nu)/2} (\sqrt{\mu_i} e_i) \otimes (\sqrt{\mu_i} e_i) \right) \phi(x) \right\|^2 \\
&= \left( \sum_{i \in I} \mu_i e_i(x)^2 (\mu_i + \lambda)^{-1+\nu} \right) \\
&= \sum_{i \in I} \mu_i^{1-\tau} (\mu_i + \lambda)^{-1+\nu} \mu_i^\tau e_i(x)^2 \\
&= \sum_{i \in I} \left( \frac{\mu_i}{\mu_i + \lambda} \right)^{1-\tau} (\mu_i + \lambda)^{\nu-\tau} \mu_i^\tau e_i(x)^2 \\
&\leq \sum_{i \in I} (\mu_i + \lambda)^{-(\tau-\nu)} \mu_i^\tau e_i(x)^2 \\
&\leq c_\tau \lambda^{-[\tau-\nu]_+} K^{2[\nu-\tau]_+}.
\end{aligned}$$

where we used $\sup |\mu_i| \leq K^2$.

$\square$

# I  Deterministic sketching results

**Lemma I.1:** *Denoting $R := I - \hat{C}_\lambda^{1/2} g_{\mathrm{KRR}}(\hat{C})\hat{C}_\lambda^{1/2}$, it holds*

$$R\hat{C}_\lambda^{1/2} = R\hat{C}_\lambda^{1/2} P_X^\perp.$$

*Proof of Lemma I.1:*  This is a direct consequence of the fact that $g_{\mathrm{KRR}}(\hat{C})\hat{C}_\lambda P_X = P_X$:

$$R\hat{C}_\lambda^{1/2} P_X = \hat{C}_\lambda^{1/2} P_X - \hat{C}_\lambda^{1/2} g_{\mathrm{KRR}}(\hat{C})\hat{C}_\lambda P_X = 0.$$

□

# J  Concentration results

## J.1  Generic concentration lemmas

All our concentration results derive from two versions of the Bernstein inequality. We first state an inequality for sums of random variables in a Hilbert space based on [70, Theorem 3.3.4], which itself derives from a result of [48].

**Lemma J.1:** *Let $(A_i)_{1 \leq i \leq n}$ be i.i.d. copies of a random variable $A$ in a separable Hilbert space $(H, \|\cdot\|)$. Assume $\mathbf{E}A = \mu$ and $\exists \sigma > 0, \exists L > 0, \forall p \geq 2, \mathbf{E}\|A - \mu\|^p \leq \frac{1}{2}p!\sigma^2 L^{p-2}$. Then for any $\delta \in ]0, 1[$ it holds:*

$$P\left[ \left\| \frac{1}{n} \sum_{i=1}^n A_i - \mu \right\| \leq \frac{2L \log(2/\delta)}{n} + \sqrt{\frac{2\sigma^2 \log(2/\delta)}{n}} \right] \geq 1 - \delta \tag{47}$$

*The assumption on the moments holds in particular when $\operatorname{ess\,sup}\|A\| \leq L/2$ and $\mathbf{E}[\|A\|^2] \leq \sigma^2$.*

*Proof of Lemma J.1:*  See proof of [11, Lemma E.3] for a precise derivation based on [70, Theorem 3.3.4]. □

We now state a version of the Bernstein concentration inequality for self-ajoint operators in operator norm, which is a restatement of [34, Lemma 24]. In the following, we denote $\mathrm{r_{eff}}(A) := \mathrm{tr}(A)/\|A\|$ the effective rank of a nonnegative definite operator $A$.

**Lemma J.2 (Bernstein for self-ajoint operators acting on a Hilbert):** *Let $H$ be a separable Hilbert space and $A_i$ be i.i.d. copies of a random variable $A$ taking values in the space of self-adjoint Hilbert-Schmidt operators on $H$. Assume $\mathbf{E}A = 0$, $\operatorname{ess\,sup}\|A\|_{\mathrm{op}} \leq c$ for some $c > 0$ (where $\|\cdot\|_{\mathrm{op}}$ denotes the operator norm) and that there exists a positive semi-definite trace class operator $V$ such that $\mathbf{E}[A^2] \preccurlyeq V$. Then for any $\delta \in ]0, 1[$ and $n \geq 1$ it holds*

$$P\left[ \left\| \frac{1}{n} \sum_{i=1}^n A_i \right\|_{\mathrm{op}} \geq \frac{2c\beta}{3n} + \sqrt{\frac{2\|V\|\beta}{n}} \right] \leq \delta \quad where \quad \beta = \log\left( \frac{4\,\mathrm{r_{eff}}(V)}{\delta} \right) \tag{48}$$

*Proof of Lemma J.2:*  See [34, Appendix B.7, Lemma 24]. □

### J.2 Applied concentration lemmas

**Lemma J.3:** *Let Assumption 4.1 hold. Let $\delta \in ]0,1[$. Then for i.i.d. samples $(x_i, y_i)_{1 \leq i \leq n}$ and any $\lambda \in ]0, \|C\|_{\mathcal{B}(\mathcal{H})}]$ it holds*

$$P\left[\left\|C_\lambda^{-1/2}(\hat{C} - C)C_\lambda^{-1/2}\right\|_{\mathcal{B}(\mathcal{H})} \geq \frac{4c_\tau\beta}{3n\lambda^\tau} + \sqrt{\frac{2c_\tau\beta}{n\lambda^\tau}}\right] \leq \delta \quad where \quad \beta = \log\left(\frac{8K^2}{\delta\lambda}\right) \quad (49)$$

*Proof of Lemma J.3:* We apply Lemma J.2 on the random variables $A_i = \xi(X_i) \otimes \xi(X_i) - C_\lambda^{-1/2}CC_\lambda^{-1/2}$ where $\xi(X_i) := C_\lambda^{-1/2}\phi(X_i)$. It holds

$$\operatorname{ess\,sup}\|A_i\|_{\mathcal{B}(\mathcal{H})} \leq 2\operatorname{ess\,sup}\|\xi(X_i)\|^2$$
$$\leq \frac{2c_\tau}{\lambda^\tau}. \quad \text{(by Lemma H.3)}$$
$$\mathbf{E}[A_i^2] = \mathbf{E}[\|\xi(X_i)\|^2\xi(X_i)\xi(X_i)^*] - (C_\lambda^{-1/2}CC_\lambda^{-1/2})^2$$
$$\preccurlyeq \mathbf{E}[\|\xi(X_i)\|^2\xi(X_i)\xi(X_i)^*]$$
$$\preccurlyeq \frac{c_\tau}{\lambda^\tau}\mathbf{E}[\xi(X_i)\xi(X_i)^*]$$
$$= \frac{c_\tau}{\lambda^\tau}CC_\lambda^{-1}$$

Thus applying Lemma J.2 with $c = \frac{2c_\tau}{\lambda^\tau}$ and $V = \frac{c_\tau}{\lambda^\tau}CC_\lambda^{-1}$, we get

$$P\left[\left\|\frac{1}{n}\sum_{i=1}^n A_i\right\| \geq \frac{4c_\tau\beta}{3n\lambda^\tau} + \sqrt{\frac{2c_\tau\beta}{\lambda^\tau n}}\right] \leq \delta \quad where \quad \beta = \log\left(\frac{8K^2}{\delta\lambda}\right) \quad (50)$$

where we used the fact that $\|CC_\lambda^{-1}\|_{\mathcal{B}(\mathcal{H})} \leq 1$ and controlled the effective rank using $\operatorname{tr}(CC_\lambda^{-1}) \leq K^2/\lambda$ and $\|CC_\lambda^{-1}\|_{\mathcal{B}(\mathcal{H})} = \|C\|_{\mathcal{B}(\mathcal{H})}/(\|C\|_{\mathcal{B}(\mathcal{H})} + 1) \geq 1/2$ because $\lambda \leq \|C\|_{\mathcal{B}(\mathcal{H})}$ by assumption. $\square$

**Lemma J.4:** *Let Assumptions 4.1 and 4.3 hold. Let $\delta \in ]0,1[$. Then for i.i.d. samples $(x_i, y_i)_{1 \leq i \leq n}$ we get*

$$P\left[\|(C - \hat{C})C_\lambda^{-1/2}\|_{\mathrm{op}} \leq \epsilon(\lambda, \delta)\right] \geq 1 - \delta \quad (51)$$

$$and \quad P\left[\|(C_{YX} - \hat{C}_{YX})C_\lambda^{-1/2}\|_{\mathrm{op}} \leq \epsilon(\lambda, \delta)\right] \geq 1 - \delta \quad (52)$$

$$where \quad \epsilon(\lambda, \delta) := \frac{2K\sqrt{c_\tau}\log(2/\delta)}{\lambda^{\tau/2}n} + \sqrt{\frac{2K^2 d_{\mathrm{eff}}(\lambda)\log(2/\delta)}{n}} \quad (53)$$

*Proof of Lemma J.4:* We first write the proof for the eq. (51). For this result, we use the fact that $\|(C - \hat{C})C_\lambda^{-1/2}\|_{\mathcal{B}(\mathcal{H})} \leq \|(C - \hat{C})C_\lambda^{-1/2}\|_{\mathrm{HS}}$ and bound the Hilbert-Schmidt norm. As $(\mathrm{HS}(\mathcal{H}), \|\cdot\|_{\mathrm{HS}(\mathcal{H})})$ is a Hilbert space, we apply Lemma J.1 on the random variables $A_i = \phi(x_i) \otimes \xi(x_i)$ where $\xi(x) = C_\lambda^{-1/2}\phi(x)$.

$$\operatorname{ess\,sup}\|A\|_{\mathrm{HS}} = \operatorname{ess\,sup}\|\phi(x)\|\|\xi(x)\|$$
$$\leq \frac{K\sqrt{c_\tau}}{\lambda^{\tau/2}} \quad \text{(by assumption 4.1 and lemma H.3)}$$
$$\mathbf{E}[\|A\|_{\mathrm{HS}}^2] = \mathbf{E}[\|\phi(x)\|^2\|\xi(x)\|^2]$$
$$\leq K^2 d_{\mathrm{eff}}(\lambda)$$

Thus applying Lemma J.1 with $L = \frac{K\sqrt{c_\tau}}{\lambda^{\tau/2}}$ and $\sigma^2 = K^2 d_{\text{eff}}(\lambda)$ gives

$$P\left[\left\|\frac{1}{n}\sum_{i=1}^n A_i - \mu\right\|_{\text{HS}} \leq \frac{2K\sqrt{c_\tau}\log(2/\delta)}{\lambda^{\tau/2}n} + \sqrt{\frac{2K^2 d_{\text{eff}}(\lambda)\log(2/\delta)}{n}}\right] \geq 1 - \delta.$$

This yields the desired result via the inequality between operator and Hilbert-Schmidt norms. For the bound eq. (52) on the cross-covariance, we take $A_i = \phi(y_i) \otimes \xi(x_i)$ but the rest of the proof is inchanged. $\qquad\square$

**Lemma J.5:** *Let Assumptions 4.1 and 4.3 hold. Let $\delta \in ]0,1[$. Then for i.i.d. samples $(x_i, y_i)_{1 \leq i \leq n}$ we get*

$$P\left[\|(C - \hat{C})C_\lambda^{-1}\|_{\text{op}} \leq \frac{2K\sqrt{c_\tau}\log(2/\delta)}{\lambda^{(\tau+1)/2}n} + \sqrt{\frac{2K^2\operatorname{tr}(C_\lambda^{-2}C)\log(2/\delta)}{n}}\right] \geq 1 - \delta. \qquad (54)$$

*Proof of Lemma J.5:* For this result, we use the fact that $\|(C - \hat{C})C_\lambda^{-1}\|_{\mathcal{B}(\mathcal{H})} \leq \|(C - \hat{C})C_\lambda^{-1}\|_{\text{HS}}$ and bound the Hilbert-Schmidt norm. As $(\text{HS}(\mathcal{H}), \|\cdot\|_{\text{HS}(\mathcal{H})})$ is a Hilbert space, we apply Lemma J.1 on the random variables $A_i = \phi(x_i) \otimes \omega(x_i)$ where $\omega(x) = C_\lambda^{-1}\phi(x)$.

$$
\begin{aligned}
\operatorname{ess\,sup}\|A\|_{\text{HS}} &= \operatorname{ess\,sup}\|\phi(x)\|\|\omega(x)\| \\
&\leq \frac{K\sqrt{c_\tau}}{\lambda^{(\tau+1)/2}} \quad \text{(by assumption 4.1 and lemma H.3)} \\
\mathbf{E}[\|A\|_{\text{HS}}^2] &= \mathbf{E}[\|\phi(x)\|^2\|\omega(x)\|^2] \\
&\leq K^2 \mathbf{E}[\operatorname{tr}(C_\lambda^{-2}\phi(x)\phi(x)^*)] \\
&= K^2 \operatorname{tr}(C_\lambda^{-2}C)
\end{aligned}
$$

Thus applying Lemma J.1 with $L = \frac{K\sqrt{c_\tau}}{\lambda^{(\tau+1)/2}}$ and $\sigma^2 = K^2 \operatorname{tr}(C_\lambda^{-2}C)$ gives

$$P\left[\left\|\frac{1}{n}\sum_{i=1}^n A_i - \mu\right\|_{\text{HS}} \leq \frac{2K\sqrt{c_\tau}\log(2/\delta)}{\lambda^{(\tau+1)/2}n} + \sqrt{\frac{2K^2\operatorname{tr}(C_\lambda^{-2}C)\log(2/\delta)}{n}}\right] \geq 1 - \delta.$$

This yields the desired result via the inequality between operator and Hilbert-Schmidt norms. $\qquad\square$

## J.3 Probabilistic inequalities

The following lemma is a restatement from [53, Lemma 6].

**Lemma J.6 (Uniform Nyström approximation):** *Let Assumption 4.1 hold. Let $P : \mathcal{H} \to \mathcal{H}$ denote the orthogonal projection on* $\operatorname{span}\left\{ \phi(\tilde{x}_j) \mid 1 \leq j \leq m \right\}$, *where the landmarks $(\tilde{x}_j)_{1 \leq j \leq m}$ are drawn uniformly without replacement from the empirical data. Then for any $\lambda \in ]0, \|C_\lambda\|_{\mathcal{B}(\mathcal{H})}]$ we have*

$$\|P^\perp C_\lambda^{1/2}\|_{\mathcal{B}(\mathcal{H})}^2 \leq 3\lambda$$

*with probability at least $1 - \delta$ provided*

$$m \geq \max(67, 5\operatorname{ess\,sup}\|C_\lambda^{-1/2}\phi(x)\|^2)\log\frac{4K^2}{\lambda\delta}.$$

## J.4 Concentration lemmas for the sketched operators

**Lemma J.7:** *It holds almost surely*

$$\|(C_{YX} - P_Y \hat{C}_{YX})C_\lambda^{-1/2}\| \leq \|P_Y^\perp C_\lambda^{1/2}\| + \|(C_{YX} - \hat{C}_{YX})C_\lambda^{-1/2}\|$$

*Proof of Lemma J.7:* It holds

$$C_{YX} - P_Y \hat{C}_{YX} = C_{YX} - P_Y C_{YX} + P_Y C_{YX} - P_Y \hat{C}_{YX}$$
$$= P_Y^\perp C_{YX} + P_Y(C_{YX} - \hat{C}_{YX})$$

Thus

$$\|(C_{YX} - P_Y \hat{C}_{YX})C_\lambda^{-1/2}\| = \|(P_Y^\perp C_{YX} + P_Y(C_{YX} - \hat{C}_{YX}))C_\lambda^{-1/2}\|$$
$$\leq \|P_Y^\perp C_{YX}C_\lambda^{-1/2}\| + \|P_Y(C_{YX} - \hat{C}_{YX})C_\lambda^{-1/2}\|$$
$$\leq \|P_Y^\perp C_\lambda^{1/2}\|\|C_\lambda^{-1/2}C_{YX}C_\lambda^{-1/2}\| + \|(C_{YX} - \hat{C}_{YX})C_\lambda^{-1/2}\|$$

Eventually it holds $\|C_\lambda^{-1/2}C_{YX}C_\lambda^{-1/2}\| \leq 1$. Indeed, as $\pi$ is invariant, it holds that

$$\|\mathcal{K}_\pi\| = \sup_{f \in L^2_\pi : \|f\|_{L^2_\pi} \leq 1} \int_x \left| \int f(y)p(x, dy) \right|^2 d\pi(x) \leq 1.$$

and denoting $\Phi_X = C^{1/2}U$ the polar decomposition of $\Phi_X$ for some partial isometry $U : L^2_\pi \to \mathcal{H}$, and using $\Phi^*_{Y|X} = \mathcal{K}_\pi \Phi^*_X$, we get

$$\|C_\lambda^{-1/2}C_{YX}C_\lambda^{-1/2}\| = \|C_\lambda^{-1/2}\Phi_{Y|X}\Phi^*_X C_\lambda^{-1/2}\|$$
$$\leq \|C_\lambda^{-1/2}C^{1/2}\|\|U\mathcal{K}^*_\pi U^*\|\|C^{1/2}C_\lambda^{-1/2}\|$$
$$\leq 1.$$

$\square$

## J.5 Concentration for mixing processes

**Lemma J.8 (Kostic et al. [29, Lemma 1]):** *Let $X$ be strictly stationary with values in a normed space $(\mathcal{X}, \|\cdot\|)$ and assume $n = 2pk$ with $p, k \in \mathbb{N}$. Let $Z_1, \ldots, Z_p$ be $p$ independent copies of $Z_1 = \sum_{i=1}^k X_i$. Then for $s > 0$:*

$$P\left[\left\|\sum_{i=1}^n X_i\right\| > s\right] \leq 2P\left[\left\|\sum_{j=1}^p Z_j\right\| > s/2\right] + 2(p-1)\beta_X(k).$$

# K  Additional experiments

In this section we extend the experiment of Figure 2 (on the Lorenz'63 dataset) to include the Nyström KRR setting. Since the KRR estimator does not use rank truncation as a regularizer, it will always output the full rank estimator. In Figure 5 we use the same settings as in the main text ($m = 250$ and $r = 50$), which lead to KRR being very close to RRR. No particular instability is encountered with these settings, and KRR is about 2x faster than RRR. In Figure 6 we increase the number of centers, without changing the amount of regularization. Now KRR is very unstable, since it needs to estimate a larger number of eigenvalues which have a very small magnitude. RRR on the other hand maintains the rank regularization and has no instability, obtaining a small performance boost thanks to the increased number of centers $m$. To improve the stability of KRR, in Figure 7 we increase the regularization from $\lambda = 10^{-4}$ to $\lambda = 5 \times 10^{-3}$. However, notice that a) the instability is not completely fixed and b) accuracy is noticeably reduced to the point where it's nearly the same as that of PCR with fewer centers.

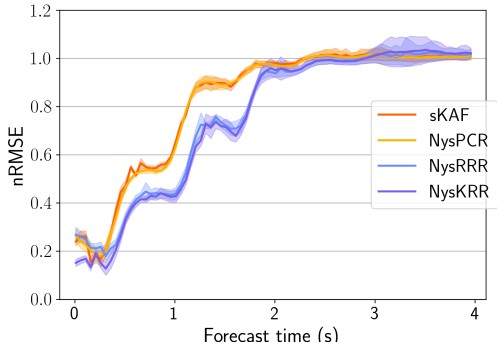

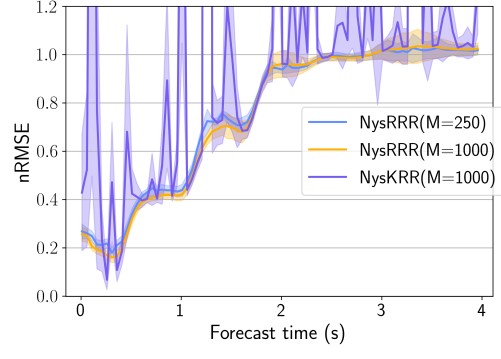

Figure 5: Forecasting error (nRMSE) versus forecast time. KRR has similar performance to RRR. We used rank $r = 50$ for scalable KAF, NysPCR, NysRRR and no rank for NysKRR. We used $m = 250$ for the Nyström estimators. For NysRRR and NysKRR we set $\lambda = 0.0001$.

Figure 6: NysKRR is very unstable. Here $r = 50$ for both NysRRR experiments, $m = 1000$ for NysKRR and NysRRR(m=1000). NysRRR(m=250) is shown for comparison with Figure 5. $\lambda = 0.0001$.

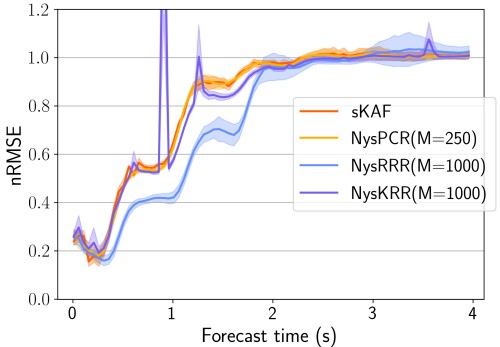

Figure 7: We increase the regularizer for NysKRR to $\lambda = 0.005$. Instability is reduced, but accuracy is also noticeably worse; in fact it is nearly equal to that of NysPCR (with fewer centers) and sKAF. Parameters for NysRRR were $r = 50, m = 1000, \lambda = 0.0001$; for NysPCR: $r = 50, m = 250$.

Connected to the instability problems, KRR is not suited for Koopman operator learning since without rank regularization, a large number of spurious eigenpairs pop up during estimation.

