# OpenReview forum: "Estimating Koopman operators with sketching to provably learn large scale dynamical systems"
_NeurIPS.cc/2023/Conference — NeurIPS 2023 poster_

### Official Review · Reviewer_8Gvs · 2023-06-20

**Soundness:** 3 good
**Presentation:** 3 good
**Contribution:** 4 excellent
**Rating:** 7
**Confidence:** 3

**Summary:**

This paper utilizes theory developed around random projections to develop fast/efficient numerical methods for approximating the Koopman mode decomposition of data sets with long-time trajectories, that additionally come with learning rates that are optimal.

**Strengths:**

1. This paper developed new fast/efficient algorithms for computing the Koopman spectra, the effectiveness of which was demonstrated on very large data sets with long-time trajectories. These include a traditional benchmark (Lorenz '63), as well as molecular data sets, which are a growing area of interest to the machine learning and Koopman communities. Additionally, the molecular data sets were of a size that could not have been studied previously.

2. This paper provided learning rates for their new algorithms, which they show to be optimal.

3. This paper discusses how different methods for computing the Koopman mode decomposition (e.g., Nystrom RRR vs. PCR) differ in their learning rate dependencies.

4. The paper was well-written.


**Weaknesses:**

1. Motivation for why very long-time trajectories of data would be needed for some systems was missing from the Introduction. This was partially discussed at the end in Sec. 5 "Molecular dynamics datasets", but developing this more is important. Additionally, the title and some places in the text describe the sketching approach as being useful for "large scale dynamical systems". As I understand it, the methods are beneficial for "long-time trajectory dynamical systems". Of course, the molecular examples shown have both large dimension and long-time trajectories, but the title and text should be corrected to accurately emphasize that it is long-time trajectory dynamical systems that this method is useful for.

2. The discussion on how Nystrom PCR and RRR differ in their learning rates was very interesting. Including more on this (maybe in table form) would be nice. Additionally, it would be helpful to see all 3 methods compared, so KRR should be included in Figure 1. Additionally, the reason for using just RRR in Figs. 3 and 4 should be noted.

3. I found it a little confusing that going from Eq. 8 to Eq. 9, the P_Y \hat{C}_{YX} P_X  became K^\dagger_{\tilde{Y}, \tilde{Y}} K_{\tilde{Y}, Y} K_{X, \tilde{X}}, but that going from Eq 10 to Eq 11 the P_Y \hat{C}_{YX}  also led to K^\dagger_{\tilde{Y}, \tilde{Y}} K_{\tilde{Y}, Y} K_{X, \tilde{X}}. I assume this has to do with the way the \dagger was absorbed into the [ \cdot ]_r?

4. How the random $\tilde{\Phi}$ are generated should be explained in the text.

Minor comments:

1. Mezic, 2005 should be cited when discussing the Koopman mode decomposition and Budisic et al. 2012 should be cited when discussing how Koopman mode decomposition has been leveraged in the past (line 95).

2. Abbreviations should be explained before using them (DMD, tICA, VAC, etc.).

3. The box plot denoting the sKAF data in Fig. 2 has a strange shape in its lower box (an inverted U). Why is this?

4. It would be helpful and appropriate to have citations when noting that "An important application of Koopman operator theory is in the analysis of molecular dynamics".

5. The colors of Fig. 3 should be explained in more detail.

6. It might be helpful to have a line across the 3 states in Fig. 4 to show how Eigenfunction 1 has linear separation between State 0 and the others, and Eigenfunction 2 has linear separation between State 1 and the others.

7. There are a few typos:
i. "allows to deploy" (line 1)
ii. "as theirs exact" (line 68)
iii. "to study large class" (line 90)
iv. "]0, 1]" (Assumption 4.3)
v. "]0, \tau]" (Assumption 4.4)


**Questions:**

To summarize, the Weakness section above,

1. Why would one want to use very long-time trajectory data when computing the Koopman mode decomposition? (You explain this some, but making it more explicit would be helpful)

2. What are the differences between KRR, PCR, and RRR in their performance on the different data sets studied (at least the Lorenz '63)?

3. How are the random $\tilde{\Phi}$ are generated?

**Limitations:**

The authors do a sufficient job explaining their limitations (the assumptions they make) and what kinds of data sets their proposed methods work best on. The only exception to this is that it should be clarified that the proposed methods improve Koopman mode decomposition estimates of long-time trajectory data and not large dimension data.

---

> ### Author Rebuttal · Authors · 2023-08-02
>
> We would like to thank the reviewer for the helpful comments. We address the main ones below, while minor comments and typos will directly be fixed in the revised version.
>
> **Motivation for long trajectories**
>
> We stress that our contribution allows to tackle large datasets, which can in practice be built using multiple trajectories in order to simply improve the overall accuracy. However, there are some applications (see molecular dynamics paragraph) where it is indeed physically necessary to record very long trajectories to capture events occurring at large scales, but that could be missed if simply subsampling the dataset. We will add a sentence to clarify this in the introduction.
>
> **Large scale vs. Long time**
>
> There is indeed a difference between the dynamical systems nomenclature, where large-scale is understood in terms of dimension of the data, and the regression setting where it usually refers to the number of data points.
> Although our contribution only focus on improving the complexity wrt the number of points, kernel methods have a linear complexity with respect to the dimension, which makes of our method an good candidate to deal with both large $n$ and $d$. We will however clarify the fact that sketching only reduces the complexity w.r.t. $n$ in order to avoid confusion.
>
> **Comparison between KRR, PCR and RRR**
>
> We added as requested KRR to the experimental results for the Lorenz' 63 dataset (see PDF).
> Explaining the behavior of Nyström KRR can be complex since it does not have a notion of rank, and will always output the full-rank estimator.
> We repeat the same experiment as in the paper, with M=250, r=50 adding the KRR estimator. In this case KRR is on par with RRR: no instability is encountered at this point, and KRR is about 2x faster than RRR. (figure 1)
> If we increase the number of centers, without increasing regularization, KRR becomes highly unstable while RRR maintains the rank regularization and hence has no problem, obtaining a small performance boost thanks to the increased M (figure 2).
> To make KRR more stable, we can increase regularization (from λ=1e-4 to λ=5e-3) and the instability is not completely fixed, while the accuracy is noticeably reduced, to the point where it's nearly the same as PCR with fewer centers (figure 3).
>
> Another reason why KRR is not a great estimator for Koopman operator learning, is that having the all $m$ components in the spectrum leads to many spurious eigenpairs popping up.
> We will condense this rather long discussion for the camera ready paper.
>
> Regarding theoretical rates of PCR and RRR, we indeed provide explicit constants in each case and discuss the role of the rank, however both estimators overall enjoy the same (optimal) rate. For this reason, it seems that adding a comparative table to the paper would not bring much. We will however stress this point in the introduction.
>
> **Estimator derivation**
>
> Yes, unfortunately the process of going from the infinite-dimensional formulation to the kernel formulation is a bit tricky. Part of the question can be answered considering $P_Y = \tilde{Φ_Y} K_{\tilde{Y}, \tilde{Y}}^† (\tilde{Φ_Y})^*$ (and similarly for X). The computations to get from the first to the second formulation involve moving part of what's in the inverse outside of it for KRR. For PCR we first simplify the $[ · ]_r^†$ expression where some of the inverses cancel out, and a similar expression outside of $[·]_r$ appears when plugging the two pieces together. The full derivation is in Appendix C.
>
> **Generation of $\tilde{Φ}_X$**
>
> The Nyström feature map comes from evaluating the normal feature map $\phi: \mathcal{X} \to \mathcal{H}$ on the so-called "Nystrom centers", which are randomly selected. We mention line 139 that we sample them uniformly, while many other strategies have been studied (e.g. leverage scores sampling, k-means).
> Something that is not mentioned (and will be added) is that the centers for X and Y are sampled in the same way (i.e. if $x_t$ is chosen as Nyström center in $\tilde{Φ_X}$ then $y_{t+1}$ is chosen as Nyström center in $\tilde{Φ_Y}$).
> We also tried empirically to take $P_X=P_Y$, which yielded a similar empirical performance.
>
> **Citations and other related work**
>
> Thanks for pointing this out, we realize that the "related work" section is missing a lot of references and we will add the two suggested references for the camera ready version.
> Similarly for the usage of Koopman operators in molecular dynamics we will reference as examples [1], [2], [3] and as a review [4]
>
> [1] "A Small Molecule Stabilizes the Disordered Native State of the Alzheimer’s Aβ Peptide", Löhr et al. (2022).
>
> [2] "Markov State Models and tICA Reveal a Nonnative Folding Nucleus in Simulations of NuG2", Schwantes et al. (2016).
>
> [3] "Slow dynamics of a protein backbone in molecular dynamics simulation revealed by time-structure based independent component analysis", Naritomi et al. (2011).
>
> [4] "Machine learning for protein folding and dynamics", Noé et al. (2020).
>
> **The colors of Fig. 3 should be explained in more detail.**
>
> We will expand the figure caption to explain that the color of the two left panels corresponds to the value of the first two eigenfunctions, evaluated at different protein conformations.
> The eigenfunctions are evaluated on the 45-d space, but the plot is overlayed on top of a 2-d subspace which is known to represent the protein dynamics well.
> The third panel shows discrete colors after clustering of the dynamics projected onto the first eigenfunctions (using PCCA+ with 3 states), showing that the obtained clusters correspond to the main areas of the Ramachandran plot.
>
> **Helpful to have a line across the 3 states in Fig. 4 to show linear separation**
>
> Thanks, we have updated the figure and the caption correspondingly (see attached PDF).
>
> **Artifact in Fig. 2**
>
> This seems to be an artifact from the matplotlib boxplot, there just is not much variance there.

---

> > ### Comment · Reviewer_8Gvs · 2023-08-12
> > **Response to authors**
> >
> > Thank you for your detailed rebuttal.
> >
> > Thank you for clarifying the confusion about large scale vs. long time. I see how this is an unavoidable issue, but I think it was explained clearly.
> >
> > These responses, as well as the reviews from the other reviewers (and the authors' responses to those reviewers) make me confident that this is a strong paper with good contributions. The changes/clarifications the authors propose to make in the revised version of the manuscript will further increase its quality.
> >
> > I will therefore increase my score.

---

### Official Review · Reviewer_jbb3 · 2023-07-05

**Soundness:** 4 excellent
**Presentation:** 4 excellent
**Contribution:** 3 good
**Rating:** 7
**Confidence:** 3

**Summary:**

This paper proposes sketching algorithms for approximating three different kernel-based estimators of Koopman operators from large scale temporal data. New non-asymptotic error bounds are derived.

**Strengths:**

- The paper leverages the concept of sketching to speed up the estimation of Koopman operators without sacrificing accuracy. The newly proposed algorithms can potentially have a high impact in several scientific disciplines where the analysis of dynamical systems is of interest.
- The authors explain and contextualize very well their contributions and the related work is comprehensive.
- The paper is very well written and organized.
- New theoretical results are rigorously derived to provide robust guarantees for the proposed methodology.
- The empirical analysis is sound and well-conducted. To showcase various aspects of the method, two distinct applications are considered. The first experiment effectively demonstrates the tradeoff between time and accuracy. The second experiment is particularly interesting as it provides a practical example where it is necessary to consider full, non-subsampled long trajectories. Furthermore, it focuses on the interpretation of dynamics and scientific discovery. It also highlights the scalability with respect to the state space dimension, which serves as motivation for the study of Koopman operator methods.

**Weaknesses:**

- While the idea of using sketching is not original, this contribution has the potential to significantly advance dynamical systems analysis by providing efficient learning algorithms for Koopman operators. Sketches for operators mapping between Hilbert spaces have recently been studied in [3]. However, the authors outline the unique aspects of their contribution and provide a clear motivation for deriving new results.

**Questions:**

- What is the complexity with respect to the dimension of the state space? Is it linear?
- Does the Markovian assumption in line 84 present a limitation for real-world applications? Is there a way to adapt Koopman operator learning (not necessarily sketched) to the non-Markovian case?

**Limitations:**

Limitations are discussed.

---

> ### Author Rebuttal · Authors · 2023-08-02
>
> We would like to thank the anonymous reviewer for their evaluation of this paper, and for their kind comments. We will address the main questions below.
>
> **Complexity with respect to the dimension of the state space?**
>
> Yes, essentially the complexity with respect to state space dimensionality is contained in the kernel function: computing an entry of the kernel typically depends linearly on the dimension of the states $d$, e.g. think about the Gaussian kernel $k(x, x') = exp(-\gamma \lVert x - x' \rVert^2)$.
>
> **Limitations of the Markovian assumption, and adaptation to the non-Markovian case**
>
> The Markovian assumption is essentially always used in molecular dynamics, where often even more restrictive assumptions are considered (such as time-homogeneity and time-reversibility). The definition of the Koopman operator could be generalized beyond this setting, for instance by considering the dynamics of the features of an history of k>1 consecutive states, and one could relatively easily extend the proposed algorithm. However the analysis in this setting would be radically different and is out of the scope of this paper; for instance the spectrum of the Koopman operator has been well studied in the Markovian setting and is known to be of key interest, while it is unclear how to generalize and leverage such a spectral analysis in a non-Markovian setting.

---

> > ### Comment · Reviewer_jbb3 · 2023-08-19
> > **Response to author rebuttal**
> >
> > Thank you for addressing my questions. I will keep my score.

---

### Official Review · Reviewer_Hh1S · 2023-07-06

**Soundness:** 3 good
**Presentation:** 2 fair
**Contribution:** 2 fair
**Rating:** 6
**Confidence:** 3

**Summary:**

This paper improves the computational efficiency of existing kernel method for Koopman operator approximation, including  KRR, PCR, RRR based on the Nystrom estimator. In this way, the authors reduce the original computational complexity from $\mathcal{O}(n^3)$ to $\mathcal{O}(n\sqrt{n})$. They provide theoretical learning bounds and several numerical studies to demonstrate the efficacy of the proposed method.

**Strengths:**

[1] The theoretical estimation of the learning bounds guarantee the performance of the proposed methods in practical scenarios.

[2] The experiments validate the efficacy of the proposed methods can scale to large dynamical systems.

**Weaknesses:**

[1] The idea of introducing random projections to the existing kernel methods is not novel enough, since the similar idea has been used in the  deep learning community, such as generative models [R1] and neural ODEs [R2].

[2] The last two experiments seem to be interesting but the authors do not describe them clearly.

[3] In line 62, the authors have typos about 'decomposition'.

__References__

[R1] Song, Y., Garg, S., Shi, J., & Ermon, S. (2020, August). Sliced score matching: A scalable approach to density and score estimation. In Uncertainty in Artificial Intelligence (pp. 574-584). PMLR.

[R2] Grathwohl, W., Chen, R. T., Bettencourt, J., Sutskever, I., & Duvenaud, D. (2018). Ffjord: Free-form continuous dynamics for scalable reversible generative models. arXiv preprint arXiv:1810.01367.

**Questions:**

[1] In the experiment of Molecular dynamics datasets, do the $\phi,\psi$ correspond to the first two eigenfunctions with largest eigenvalues of the approximated Koopman matrix?  And why not demonstrate the predictive performance of the proposed methods in the original 45-D state space in this task?

**Limitations:**

The authors have not provided experiments to demonstrate the efficacy of the proposed methods with the non-i.i.d. data, but the current work is complete anyway.

---

> ### Author Rebuttal · Authors · 2023-08-02
>
> We would first of all like to thank the reviewer for taking the time to read this paper and helping us improve its current state. We will address the main points raised below.
>
> **Novelty of random projections & comparison to random projections for generative models and neural ODEs.**
>
> Random projections are indeed an old idea, since they provide a computationally efficient way of performing dimensionality reduction. They have been used in many different areas of computer science, including more recently deep learning. In kernel methods, random projections often go under the name of the Nyström method when they can be interpreted as subsampling operations on the kernel matrix.
> However, there are far too many instances of random projections in machine learning to mention them all, or even to give a partial overview. For this reason we only cite the ones which are more relevant to the models being considered, similarly to also for example [R1] which does not cite the Nyström method in kernel ridge regression.
>
> **Clarity of last two experiments**
>
> For a description of the experiment on the alanine dipeptide please see the next point. Regarding the last experiment, we will try to explain it better here. Please let us know if some aspects remain unclear.
> After learning the Koopman operator with the Nystrom RRR model on a training set, we compute the first three eigenfunctions on the test dataset.
> We then train a clustering model on top of the eigenfunctions (PCCA+ model) with three states.
> In the plot, for each eigenfunction-state pair, we have time along the trajectory on the x-axis, the eigenfunction value on the y-axis, and color indicating the probability of that particular state. So eigenfunction 1 clearly distinguishes between states 1 and 2, while eigenfunction 2 clearly distinguishes between states 0 and 2.
> Finally we plot some representative structures for each state showing that indeed we have identified three different conformations that the protein can take throughout the trajectory, distinguishing them in a very high-dimensional input state space.
>
> **Meaning of $\phi, \psi$ in the alanine dipeptide experiment. Why not predictive performance on original state space?**
>
> The alanine dipeptide can be described in two ways: either through a 45d state (pairwise distances between heavy atoms), or through a 2d state (two angles between backbone atoms, called $\phi$ and $\psi$).
> It is well known that the important dynamics of the protein (changes in the conformation which are stable over time) can be characterized fully from the 2d state. In particular there are 2 transitions between the left and right parts of the plot and top and bottom parts.
> Through our experiment we wish to show that even though we learn the Koopman operator on the larger state, the first two eigenfunctions can capture the stable transitions of the protein well.
> To show this we can overlay the eigenfunction values on top of the 2d-state plot, and verify that the eigenfunction changes are highly correlated with changes in the angles $\phi$ and $\psi$.
> On this task, predictive performance (i.e. forcasting $x_{t+1}$ from $x_t$) is not a common metric to use. In fact the main reason for learning a dynamical system through the Koopman operator is to obtain the spectrum. If one were not interested in that, and just in forecasting, there are lots of other ways to tackle the problem. This is the reason behind our focus on the eigenfunctions here.

---

> > ### Comment · Reviewer_Hh1S · 2023-08-19
> > **Review to the rebuttal**
> >
> > Thanks for the detailed rebuttal, after reading the general response as well as the other reviews, I am happy to amend my score of the paper.

---

### Official Review · Reviewer_FXhE · 2023-07-06

**Soundness:** 3 good
**Presentation:** 3 good
**Contribution:** 2 fair
**Rating:** 6
**Confidence:** 3

**Summary:**

The paper talks about estimating Koopman operators. Sketching (random projections), specifically the Nyström method is used as a computational tool to make things tractable. Furthermore, it is shown that these approximates have the same convergence rate as their slower counter parts. Further error bounds are provided in terms of the operator norm.

**Strengths:**

The authors do a good job in positing the current work within the literature which is very wide otherwise.

The flow of the paper is very good and reads well.

Assumptions in Section 4 are clearly mentioned before stating the main result.

I appreciate authors commenting on the non iid case, apart from detailed analysis of the iid scenario.


**Weaknesses:**

I view the papers contributions as two fold.

The first part is the use of sketching for approximation of the three scenarios of KRR, PCR, RRR considered. In this case (as authors have acknowledged, references [3] and [6] therein), there are  other works who have pointed this out.

Again authors mention the special case of their sketched PCR as kernel DMD. These make the contributions of the current work from this point of view a bit restricted. Having said that, I greatly appreciate authors mentioning them in the paper!

The second is in terms of learning bounds via operator norm, where I think the contribution is solid.

However, it is hard to have empirical evidence to support this and is understandable. It would be good to have some discussion around this, as Koopman operators are heavily used in practical settings and having some evidence would greatly improve the impact of the paper over it's current form.

**Questions:**

1) What is $A^\ast$ is line 104? The equation 5 just above has only $A$, are they the same?
2) In experimental evaluation, only PCR and RRR are considered, what about KRR ?

**Limitations:**

I don't see a direct potential negative social impact.

---

> ### Author Rebuttal · Authors · 2023-08-02
>
> We thank the reviewer for their helpful comments, which we address below.
>
> **Novelty and comparison with refs. [3], [6] and kernel DMD.**
>
> References [3] and [6] deal with the Nystrom KRR model, with [3] being more concerned with theoretical considerations (in a slightly different setting from us: [3] bounds convergence rates in Hilbert-Schmidt norm, while our work is in operator norm) and [6] with the physical interpretation of the resulting model, without providing theoretical guarantees.
> Our contributions are a generalization of these works to two different estimators: PCR and RRR. We argue that these estimators are particularly useful for Koopman operator learning where the desired rank $r$ can be much smaller than the number of centers $m$ (see for example our last experiment), and it is only necessary to estimate the first $r$ components of the Koopman operator's spectrum. Using rank-reduced estimators reduces the risk of spectrum pollution and (at least in the case of $r \ll m$) reduces the computational complexity).
>
> In fact, kernel PCR is equivalent to the kernel DMD algorithm. Compared to kernel DMD, our contribution is the introduction of sketching, which greatly improves the computational complexity making long trajectories (or more generally a large total number of time-steps) easy to handle.
>
> **Empirical evidence to support the learning bounds.**
>
> With regards to a qualitative demonstration of the learning bounds, we can use the l63 system as an example with relatively small amounts of noise. In Fig. 4 of the rebuttal pdf (and correspondingly in the paper) we show  that -- as the number of training samples increases the test error decreases as $\sim n^{-1/2}$. While we increase $n$, we must also increase $m$ to satisfy the assumptions (in particular, we set $m = 5\sqrt{n}$).
> We perform a qualitative fit of the test error to set two parameters $a$ and $b$ which contain the unknown constants in our theorems ($\mathcal{E}(\hat{A}) \sim \frac{a}{\sqrt{n}} + b$), to show that the error approximately follows the expected rate.
> Unfortunately this experiment is too coarse to distinguish between the rates of PCR and RRR, which are also influenced by hyperparameters such as $\lambda$, and constants depending on the particular dataset. Note that KRR becomes unstable as $n$ (and correspondingly $m$) increases, since it needs to estimate more and more eigenvalues.
>
> **What is A\* line 104?**
>
> Yes, they are the same operator. $A^*$ denotes the adjoint of $A$. Due to the way we formulate the problem in eq. 5 $A$ is the estimator of the adjoint of the Koopman operator, or equivalently $A^*$ is the estimator of the Koopman operator itself. If the passage is a bit unclear we could change the text to say:
>
>     The operator $A^*$ should thus be understood as an estimator of the adjoint of the Koopman operator ...
>
>
> **In experimental evaluation, only PCR and RRR are considered, what about KRR ?**
>
> We will add KRR to the Lorenz '63 experiment. Please see the "Author rebuttal" (and the accompanying figures in the PDF) for how this would look like, and some motivation for why this was not done in the first place.

---

> > ### Comment · Reviewer_FXhE · 2023-08-18
> >
> > Thank you for the rebuttal. Please clarify the adjoint notation, it will be helpful. I appreciate the experiments added and am increasing my score.

---

### Author Rebuttal · Authors · 2023-08-02

We thank all the anonymous reviewers for their thoughtful comments.
We provide here a pdf file with additional experimental results, and address directly the comments of the reviewers in the individual rebuttals.

In particular, the experiments concern the addition of Nystrom KRR to a comparison with the other two (PCR and RRR) estimators, which was aked for by two reviewers. We repeat here the comments made on the rebuttal for Reviewer 8Gvs since they may be of interest to the other Reviewers.

**Comparison between KRR, PCR and RRR**

We added as requested KRR to the experimental results for the Lorenz' 63 dataset (see PDF). Explaining the behavior of Nyström KRR can be complex since it does not have a notion of rank, and will always output the full-rank estimator. We repeat the same experiment as in the paper, with M=250, r=50 adding the KRR estimator. In this case KRR is on par with RRR: no instability is encountered at this point, and KRR is about 2x faster than RRR. (figure 1) If we increase the number of centers, without increasing regularization, KRR becomes highly unstable while RRR maintains the rank regularization and hence has no problem, obtaining a small performance boost thanks to the increased M (figure 2). To make KRR more stable, we can increase regularization (from λ=1e-4 to λ=5e-3) and the instability is not completely fixed, while the accuracy is noticeably reduced, to the point where it's nearly the same as PCR with fewer centers (figure 3).

Another reason why KRR is not a great estimator for Koopman operator learning, is that having the all components in the spectrum leads to many spurious eigenpairs popping up. We will condense this rather long discussion for the camera ready paper.

---

### Decision · Program_Chairs · 2023-09-21

**Decision:**

Accept (poster)

**Comment:**

The paper presents novel "sketched" estimators for learning Koopman operators.. The authors place their work within the wider literature effectively, which was well received by the reviewers. Kernel-based Koopman operator estimators are commonly used to predict and analyze complex dynamical systems, but their application to lengthy trajectories leads to computational challenges. This paper proposes to use random projections (sketching) to address this issue.

Beyond its theoretical contribution, the paper demonstrates practical utility through a series of well executed simulations. The empirical evaluation involved diverse settings, highlighting the trade-off between time and accuracy, and included a case where non-subsampled long trajectories were necessary.

The proposed methodology is convincing - a combination that could be impactful across fields where the analysis of dynamical systems is central. Given its potential impact in the field of large-scale dynamical systems, this paper is unanimously recommended for acceptance.